# Out-of-Distribution Generalization Analysis via Influence Function

## Abstract

The mismatch between training and target data is one major challenge for current machine learning systems. When training data is collected from multiple domains and the target domains include all training domains and other new domains, we are facing an Out-of-Distribution (OOD) generalization problem that aims to find a model with the best OOD accuracy. One of the definitions of OOD accuracy is worst-domain accuracy. In general, the set of target domains is unknown, and the worst over target domains may be unseen when the number of observed domains is limited. In this paper, we show that the worst accuracy over the observed domains may dramatically fail to identify the OOD accuracy. To this end, we introduce Influence Function, a classical tool from robust statistics, into the OOD generalization problem and suggest the variance of influence function to monitor the stability of a model on training domains. We show that the accuracy on test domains and the proposed index together can help us discern whether OOD algorithms are needed and whether a model achieves good OOD generalization.

## 1 Introduction

Most machine learning systems assume both training and test data are independently and identically distributed, which does not always hold in practice (Bengio et al. (2019)). Consequently, its performance is often greatly degraded when the test data is from a different domain (distribution). A classical example is the problem to identify cows and camels (Beery et al. (2018)), where the empirical risk minimization (ERM, Vapnik (1992)) may classify images by background color instead of object shape. As a result, when the test domain is "out-of-distribution" (OOD), e.g. when the background color is changed, its performance will drop significantly. The OOD generalization is to obtain a robust predictor against this distribution shift.

Suppose that we have training data collected from $m$ domains:

$$\mathbb{S} = \{\mathbb{S}^e : e \in \mathcal{E}_{tr}, |\mathcal{E}_{tr}| = m\}, \quad \mathbb{S}^e = \{\boldsymbol{z}_1^e, \boldsymbol{z}_2^e, \ldots, \boldsymbol{z}_{n^e}^e\} \text{ with } \boldsymbol{z}_i^e \sim P^e, \tag{1}$$

where $P^e$ is the distribution corresponding to domain $e$, $\mathcal{E}_{tr}$ is the set of *all available domains, including validation domains*, and $\boldsymbol{z}_i^e$ is a data point. The OOD problem we considered is to find a model $f_{\text{OOD}}$ such that

$$f_{\text{OOD}} = \arg\min_f \sup_{P^e \in \mathcal{E}_{all}} \ell(f, P^e), \tag{2}$$

where $\mathcal{E}_{all}$ is the set of all target domains and $\ell(f, P^e)$ is the expected loss of $f$ on the domain $P^e$. Recent algorithms address this OOD problem by recovering invariant (causal) features and build the optimal model on top of these features, such as Invariant Risk Minimization (IRM, Arjovsky et al. (2019)), Risk Extrapolation (REx, Krueger et al. (2020)), Group Distributionally Robust Optimization (gDRO, Sagawa et al. (2019)) and Inter-domain Mixup (Mixup, Xu et al. (2020); Yan et al. (2020); Wang et al. (2020)). Most works evaluate on Colored MNIST (see 5.1 for details) where we can directly obtain the worst domain accuracy over $\mathcal{E}_{all}$. Gulrajani & Lopez-Paz (2020) has assembled many algorithms and multi-domain datasets, and finds that OOD algorithms can't outperform ERM in some domain generalization tasks (Gulrajani & Lopez-Paz (2020)), e.g. VLCS (Torralba & Efros (2011)) and PACS (Li et al. (2017)). This is not surprising, since these tasks only require high performance on certain domains, while an OOD algorithm is expected to learn truly invariant

features and be excellent on a large set of target domains $\mathcal{E}_{all}$. This phenomenon is described as "accuracy-vs-invariance trade-off" in Akuzawa et al. (2019).

Two questions arise in the min-max problem (2). First, previous works assume that there is sufficient diversity among the domains in $\mathcal{E}_{all}$. Thus the supremacy of $\ell(f, P^e)$ may be much larger than the average, which implies that ERM may fail to discover $f_{OOD}$. But in reality, we do not know whether it is true. If not, the distribution of $\ell(f, P^e)$ is concentrated on the expectation of $\ell(f, P^e)$, and ERM is sufficient to find an invariant model for $\mathcal{E}_{all}$. Therefore, we call for a method to judge whether an OOD algorithm is needed. Second, how to judge a model's OOD performance? Traditionally, we consider test domains $\mathcal{E}_{test} \subset \mathcal{E}_{tr}$ and use the worst-domain accuracy over $\mathcal{E}_{test}$ (which we call test accuracy) to approximate the OOD accuracy. However, test accuracy is a biased estimate of the OOD accuracy unless $\mathcal{E}_{tr}$ is closed to $\mathcal{E}_{all}$. More seriously, It may be irrelevant or even *negatively correlated* to the OOD accuracy. This phenomenon is not uncommon, especially when there are features virtually spurious in $\mathcal{E}_{all}$ but show a strong correlation to the target in $\mathcal{E}_{tr}$.

We give a toy example in Colored MNIST when the test accuracy fails to approximate the OOD accuracy. For more details, please refer to Section 5.1 and Appendix A.4. We choose three domains from Colored MNIST and use cross-validation (Gulrajani & Lopez-Paz (2020)) to select models, i.e. we take turns to select a domain $S \in \mathcal{E}_{tr}$ as the test domain and train on the rest, and select the model with max average test accuracy. Figure 1 shows the comparison between ERM and IRM. One can find that no matter which domain is the test domain, ERM model uniformly outperforms IRM model on the test domain. However, IRM model achieves consistently better OOD accuracy. Shortcomings of the test accuracy here are obvious, regardless of whether cross-validation is used. In short, the naive use of the test accuracy may result in a non-OOD model.

To address this obstacle, we hope to find a metric that correlates better with model's OOD property, *even when $\mathcal{E}_{tr}$ is much smaller than $\mathcal{E}_{all}$ and the "worst" domain remains unknown*. Without any assumption to $\mathcal{E}_{all}$, our goal is unrealistic. Therefore, we assume that features that are invariant across $\mathcal{E}_{tr}$ should also be across $\mathcal{E}_{all}$. This assumption is necessary. Otherwise, the only thing we can do is to collect more domains. Therefore, we need to focus on what features the model has learnt. Specifically, we want to check whether the model learns invariant features and avoid varying features.

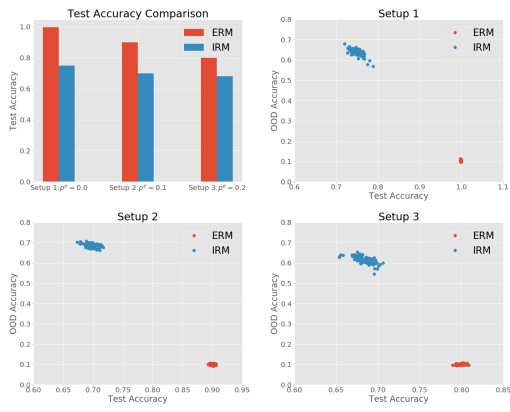

The influence function (Cook & Weisberg (1980)) can serve our purpose. Influence function was proposed to measures the parameter change when a data point is removed or upweighted by a small perturbation (details in 3.2). When modified

Figure 1: Experiments in Colored MNIST to show test accuracy is not enough to reflect a model's OOD accuracy. The top left penal shows the test accuracy of ERM and IRM. The other three panels present the relationship between test accuracy (x-axis) and OOD accuracy (y-axis) in three setups.

it to domain-level, it measures the influence of a domain instead of a data point on the model. Note that we are not emulating the changes of the parameter when a domain is removed. Instead, we are exactly caring about upweighting the domain by $\delta \to 0^+$ (will be specified later). Base on this, the variance of influence function allows us to measure OOD property and solve the obstacle.

**Contributions** we summarize our contributions here: (i) We introduce influence function to domain-level and propose index $\mathcal{V}_{\gamma|\theta}$ (formula 6) based on influence function of the model $f_{\theta}$. Our index can measure the OOD extent of available domains, i.e. how different these domains (distributions) are. This measurement provides a basis for whether to adopt an OOD algorithm and to collect more diverse domains. See Section 4.1 and Section 5.1.1 for details. (ii) We point out that the

proposed index $\mathcal{V}_{\gamma|\boldsymbol{\theta}}$ can solve the weakness of test accuracy. Specifically, under most OOD generalization problems, using test accuracy and our index together, we can discern the OOD property of a model. See Section 4.2 for details. (iii) We propose to use only a small but important part of the model to calculate the influence function. This overcomes the huge computation cost of solving the inverse of Hessian. It is not merely for calculation efficiency and accuracy, but it coincides with our understanding that only these parameters capture what features a model has learnt (Section 4.3).

We organize our paper as follows: Section 2 reviews related works and Section 3 introduces the preliminaries of OOD methods and influence function. Section 4 presents our proposal and detailed analysis. Section 5 shows our experiments. The conclusion is given in Section 6.

## 2 RELATED WORK

The mismatch between the development dataset and the target domain is one major challenge in machine learning (Castro et al. (2020); Kuang et al. (2020)). Many works assume that the ground truth can be represented by a causal Direct Acyclic Graph (DAG), and they use the DAG structure to discuss the worst-domain performance (Rojas-Carulla et al. (2018); Peters et al. (2016); Subbaswamy et al. (2019); Bühlmann et al. (2020); Magliacane et al. (2018)). All these works employ multiple domain data and causal assumptions to discover the parents of the target variable. Rojas-Carulla et al. (2018) and Magliacane et al. (2018) also apply this idea to Domain Generalization and Multi-Task Learning setting. Starting from multiple domain data rather than model assumptions, Arjovsky et al. (2019) proposes Invariant Risk Minimization (IRM) to extract causal (invariant) features and learn invariant optimal predictor on the top of the causal features. It analyzes the generalization properties of IRM from the view of sufficient dimension reduction (Cook (2009); Cook et al. (2002)). Ahuja et al. (2020) considers IRM as finding the Nash equilibrium of an ensemble game among several domains and develops a simple training algorithm. Krueger et al. (2020) derives the Risk Extrapolation (REx) to extract invariant features and further derives a practical objective function via variance penalization. Xie et al. (2020) employs a framework from distributional robustness to interpret the benefit of REx comparing to robust optimization (Ben-Tal et al. (2009); Bagnell (2005)). Besides, Adversarial Domain Adaption (Li et al. (2018); Koyama & Yamaguchi (2020)) uses discriminator to look for features that are independent of domains and uses these features for further prediction.

Influence function is a classic method from the robust statistics literature (Robins et al. (2008; 2017); Van der Laan et al. (2003); Tsiatis (2007)). It can be used to track the impact of a training sample on the prediction. Koh & Liang (2017) proposes a second-order optimization technique to approximate the influence function. They verify their method with different assumptions on the empirical risk ranging from being strictly convex and twice-differentiable to non-convex and non-differentiable losses. Koh et al. (2019) also estimates the effect of removing a subgroup of training points via influence function. They find out that the approximation computed by the influence function is correlated with the actual effect. Influence function has been used in many machine learning tasks. Cheng et al. (2019) proposes an explanation method, Fast Influence Analysis, that employs influence function on Latent Factor Model to solve the lack of interpretability of the collaborative filtering approaches for recommender systems. Cohen et al. (2020) uses influence function to detect adversarial attacks. Ting & Brochu (2018) proposes an asymptotically optimal sampling method via an asymptotically linear estimator and the associated influence function. Alaa & Van Der Schaar (2019) develops a model validation procedure that estimates the estimation error of causal inference methods. Besides, Fang et al. (2020) leverages influence function to select a subset of normal users who are influential to the recommendations.

## 3 PRELIMINARIES

### 3.1 ERM, IRM AND REX

In this section, we give some notations and introduce some recent OOD methods. Recall the multiple domain setup (1) and OOD problem (2). For a domain $P^e$ and a hypothetical model $f$, the population loss is $\ell(f, P^e) = \mathbb{E}_{\mathbf{z} \sim P^e}[L(f, \mathbf{z})]$ where $L(f, \mathbf{z})$ is the loss function on $\mathbf{z}$. The empirical loss, which is the objective of ERM, is $\ell(f, \mathbb{S}) = (1/m) \sum_{e \in \mathcal{E}_{tr}} \ell(f, \mathbb{S}^e)$ with $\ell(f, \mathbb{S}^e) = (1/n) \sum_{i=1}^{n} L(f, \mathbf{z}_i^e)$.

Recent OOD methods propose some novel regularized objective functions in the form:

$$\mathcal{L}(f, \mathbb{S}) = \ell(f, \mathbb{S}) + \lambda R(f, \mathbb{S}) \tag{3}$$

to discover $f_{\text{OOD}}$ in (2). Here $R(f, \mathbb{S})$ is a regularization term and $\lambda$ is the tuning parameter which controls the degree of penalty. Note that ERM is a special case by setting $\lambda = 0$. For simplicity, we will use $\mathcal{L}(f, \mathbb{S})$ to represent the total loss in case of no ambiguity. Arjovsky et al. (2019) focuses on the stability of $f_{\text{OOD}}$ and considers the IRM regularization:

$$R(f, \mathbb{S}) = \sum_{e \in \mathcal{E}_{tr}} \left\| \nabla_w \ell\big(wf), \mathbb{S}^e\big)\big|_{w=1.0} \right\|^2 \tag{4}$$

where $w$ is a scalar and fixed "dummy" classifier. Arjovsky et al. (2019) shows that the scalar fixed classifier $w$ is sufficient to monitor invariance and responds to the idealistic IRM problem which decomposes the entire predictor into data representation and one shared optimal top classifier for all training domains. On the other hand, Krueger et al. (2020) encourages the uniform performance of $f_{\text{OOD}}$ and proposes the V-REx penalty:

$$R(f, \mathbb{S}) = \sum_{e \in \mathcal{E}_{tr}} \left( \ell(f, \mathbb{S}^e) - \ell(f, \mathbb{S}) \right)^2.$$

Krueger et al. (2020) derives the invariant prediction by the robustness to spurious features and figure out that REx is more robust than group distributional robustness (Sagawa et al. (2019)). In this work, we also decompose the entire predictor into a feature extractor and a classifier on the top of the learnt features. As we will see, different from Arjovsky et al. (2019) and Krueger et al. (2020), we directly monitor the invariance of the top model.

## 3.2 Influence function and group effect

Consider a parametric hypothesis $f = f_{\boldsymbol{\theta}}$ and the corresponding solution: $\hat{\boldsymbol{\theta}} = \arg\min_{\boldsymbol{\theta}} \mathcal{L}(f_{\boldsymbol{\theta}}, \mathbb{S})$. By a quadratic approximation of $\mathcal{L}(f_{\boldsymbol{\theta}}, \mathbb{S})$ around $\hat{\boldsymbol{\theta}}$, the influence function takes the form

$$\mathcal{IF}(\hat{\boldsymbol{\theta}}, \boldsymbol{z}) = -\boldsymbol{H}_{\hat{\boldsymbol{\theta}}}^{-1} \nabla_{\boldsymbol{\theta}} L(f_{\hat{\boldsymbol{\theta}}}, \boldsymbol{z}) \quad \text{with} \quad \boldsymbol{H}_{\hat{\boldsymbol{\theta}}} = \nabla_{\boldsymbol{\theta}}^2 \mathcal{L}(f_{\hat{\boldsymbol{\theta}}}, \mathbb{S}).$$

When the sample size of $\mathbb{S}$ is sufficiently large, the parameter change due to removing a data point $z$ can be approximated by $-\mathcal{I}(z) / \sum_{e \in \mathcal{E}_{tr}} |\mathbb{S}^e|$ without retraining the model. Here $|\mathbb{S}^e| = n^e$ stands for the cardinal of the set $\mathbb{S}^e$. Furthermore, Koh et al. (2019) shows that the influence function can also predict the effects of large groups of training points (i.e. $\mathcal{Z} = \{z_1, ..., z_k\}$), although there are significant changes in the model. The parameter change due to removing the group can be approximated by

$$\mathcal{IF}(\hat{\boldsymbol{\theta}}, \mathcal{Z}) = -\boldsymbol{H}_{\hat{\boldsymbol{\theta}}}^{-1} \nabla_{\boldsymbol{\theta}} \frac{1}{|\mathcal{Z}|} \sum_{z \in \mathcal{Z}} L(f_{\hat{\boldsymbol{\theta}}}, z).$$

Motivated by the work of Koh et al. (2019), we introduce influence function to OOD problem to address our obstacles.

# 4 Methodology

## 4.1 Influence of domains

We decompose a parametric hypothesis $f_{\boldsymbol{\theta}}(x)$ into a top model $g$ and a feature extractor $\Phi$, i.e. $f_{\boldsymbol{\theta}}(x) = g(\Phi(\boldsymbol{x}, \boldsymbol{\beta}), \boldsymbol{\gamma})$ and $\boldsymbol{\theta} = (\boldsymbol{\gamma}, \boldsymbol{\beta})$. Such decomposition coincides the understanding of most DNN, i.e. a DNN extracts the features and build a top model based on the extracted features. When upweighting a domain $e$ by a small perturbation $\delta$, we do not upweight the regularized term, i.e.

$$\mathcal{L}_+(\boldsymbol{\theta}, \mathbb{S}, \delta) = \mathcal{L}(\boldsymbol{\theta}, \mathbb{S}) + \delta \cdot \ell(f, \mathbb{S}^e),$$

since the stability across different domains, which is encouraged by the regularization, should not depend on the sample size of a domain. For a learnt model $f_{\hat{\boldsymbol{\theta}}}$, fixing the feature extractor $\Phi$, i.e. fixing $\boldsymbol{\beta} = \hat{\boldsymbol{\beta}}$, the change of top model $g$ caused by upweighting the domain is

$$\mathcal{IF}(\hat{\boldsymbol{\gamma}}, \mathbb{S}^e | \hat{\boldsymbol{\theta}}) := \lim_{\delta \to 0^+} \frac{\Delta \boldsymbol{\theta}}{\delta} = -\boldsymbol{H}_{\hat{\boldsymbol{\gamma}}}^{-1} \nabla_{\boldsymbol{\gamma}} \ell(f_{\hat{\boldsymbol{\theta}}}, \mathbb{S}^e), \quad e \in \mathcal{E}_{tr}. \tag{5}$$

Here $\boldsymbol{H}_{\hat{\gamma}} = \nabla_{\hat{\gamma}}^2 \mathcal{L}(f_{\hat{\boldsymbol{\theta}}}, \mathbb{S})$, and we assume $\mathcal{L}$ is twice-differentiable in $\boldsymbol{\gamma}$. Please see Appendix A.3 for detailed derivation and why $\boldsymbol{\beta}$ should be fixed. For a regularized method, e.g. IRM and REx, the influence of their regularized term is reflected in $\boldsymbol{H}$ and in learnt model $f_{\hat{\boldsymbol{\theta}}}$. As mentioned above, $\mathcal{IF}(\hat{\boldsymbol{\gamma}}, \mathbb{S}^e | \hat{\boldsymbol{\theta}})$ measures change of model caused by upweighting domain $e$. Therefore, if $g(\Phi, \hat{\boldsymbol{\gamma}})$ is invariant across domains, the entire model $f_{\hat{\boldsymbol{\theta}}}$ treats all domains equally. As a result, a small perturbation on different domains should cause *the same model change*. This leads to our proposal.

## 4.2 PROPOSED INDEX AND ITS UTILITY

On basis of the domain-level influence function $\mathcal{IF}(\hat{\boldsymbol{\gamma}}, \mathbb{S}^e | \hat{\boldsymbol{\theta}})$, we propose our index to measure the fluctuation of the parameter change when different domains are upweighted:

$$\mathcal{V}_{\hat{\gamma}|\hat{\boldsymbol{\theta}}} := \ln \left( \| \mathrm{Cov}_{e \in \mathcal{E}_{tr}} \big( \mathcal{IF}(\hat{\boldsymbol{\gamma}}, \mathbb{S}^e | \hat{\boldsymbol{\theta}}) \big) \|_2 \right). \tag{6}$$

Here $\| \cdot \|_2$ is the 2-norm for matrix, i.e. the largest eigenvalue of the matrix, $\mathrm{Cov}_{e \in \mathcal{E}_{tr}}(\cdot)$ refers to the covariance matrix of the domain-level influence function over $\mathcal{E}_{tr}$ and $\ln(\cdot)$ is a nonlinear transformation that works well in practice.

**OOD Model** Under the OOD problem in (2), a good OOD model should (i) learn invariant and useful features; (ii) avoid spurious and varying features. Learning useful and invariant features means the model should have high accuracy over a set of test domains $\mathcal{E}_{test}$, no matter which test domain it is. In turn, high accuracy over $\mathcal{E}_{test}$ also means the model truly learns some useful features for the test domains. However, this is not enough, since we do not know whether the useful features are invariant features across $\mathcal{E}_{all}$ or just spurious features on $\mathcal{E}_{test}$. On the other hand, avoiding varying features means that different domains are *actually the same* to the learnt model, so according to the arguments in Section 4.1, $\mathcal{V}_{\gamma|\boldsymbol{\theta}}$ should be small. Combined this, we derive our proposal: *if a learnt model $f_{\hat{\boldsymbol{\theta}}}$ manage to simultaneously achieve small $\mathcal{V}_{\hat{\gamma}|\hat{\boldsymbol{\theta}}}$ and high accuracy over $\mathcal{E}_{test}$, it should have good OOD accuracy.* We prove our proposal in a simple but illuminating case, and we conduct various experiments (Section 5) to support our proposal. Several issues should be clarified. First, not all OOD problems demand models to learn invariant features. For example, the set of all target domains is small such that the varying features are always strongly correlated to the labels, or the objective is the mean of the accuracy over $\mathcal{E}_{all}$ rather than the worst-domain accuracy. But to our concern, we regard the OOD problem in (2) as a bridge to causal discover. Thus the set of the target domains is large, and the "weak" OOD problems are out of our consideration. To a large extent, invariant features are still the major target and our proposal is still a good criterion to model's OOD property. Second, we admit that the gap between being stable in $\mathcal{E}_{tr}$ (small $\mathcal{V}_{\gamma|\boldsymbol{\theta}}$) and avoiding all spurious features on $\mathcal{E}_{all}$ does exist. However, to our knowledge, for features that are varying in $\mathcal{E}_{all}$ but are invariant in $\mathcal{E}_{tr}$, demanding a model to avoid them is somehow unrealistic. Therefore, we make a step forward that we measure whether the learnt model successfully avoids features that vary across $\mathcal{E}_{tr}$. We leave index about varying features over $\mathcal{E}_{all}$ in our future work.

**The Shuffle $\mathcal{V}_{\gamma|\boldsymbol{\beta}}$** As mentioned above, smaller metric $\mathcal{V}_{\gamma|\boldsymbol{\theta}}$ means strong stablility across $\mathcal{E}_{tr}$, and hence should have better OOD accuracy. However, the proposed metric depends on the dataset $\mathbb{S}$ and the learnt model $f_{\hat{\boldsymbol{\theta}}}$. Therefore, there is no uniform baseline to check whether the metric is "small" enough. To this end, we propose a baseline value of the proposed metric by shuffling the multi-domain data. Consider pooling all data points in $\mathbb{S}$ and randomly redistributed to $m$ new synthetic domains $\{\tilde{\mathbb{S}}^1, \tilde{\mathbb{S}}^2, ..., \tilde{\mathbb{S}}^m\} := \tilde{\mathbb{S}}$. We compute *the shuffle version* of $\mathcal{V}_{\gamma|\boldsymbol{\theta}}$ for a learnt model $f_{\hat{\boldsymbol{\theta}}}$ over the shuffled data $\tilde{\mathbb{S}}$:

$$\tilde{\mathcal{V}}_{\hat{\gamma}|\hat{\boldsymbol{\theta}}} := \ln \left( \| \mathrm{Cov}_{e \in \mathcal{E}_{tr}} \big( \mathcal{IF}(\hat{\boldsymbol{\gamma}}, \mathbb{S}^e | \hat{\boldsymbol{\theta}}) \big) \|_2 \right). \tag{7}$$

and denote the standard version and shuffle version of the metric as $\mathcal{V}_{\hat{\gamma}|\hat{\boldsymbol{\theta}}}$ and $\tilde{\mathcal{V}}_{\hat{\gamma}|\hat{\boldsymbol{\theta}}}$ respectively. For any algorithm that obtains relatively good test accuracy, if $\mathcal{V}_{\hat{\gamma}|\hat{\boldsymbol{\theta}}}$ is much larger than $\tilde{\mathcal{V}}_{\hat{\gamma}|\hat{\boldsymbol{\theta}}}$, $f_{\hat{\boldsymbol{\theta}}}$ has learnt features that vary across $e \in \mathcal{E}_{tr}$, and cannot treat domains in $\mathcal{E}_{tr}$ equally. This implies that $f_{\hat{\boldsymbol{\theta}}}$ may not be an invariant predictor over $\mathcal{E}_{all}$. Otherwise, if the two values are similar, the model has avoided varying features in $\mathcal{E}_{tr}$ and maybe invariant across $\mathcal{E}_{tr}$. Therefore, either the model capture the invariance over the diverse domains, or the domains are not diverse at all. Note that

this process is suitable for any algorithm, hence providing a baseline to see whether $\mathcal{V}_{\hat{\gamma}|\hat{\theta}}$ is small. Here we also obtain a method to judge whether an OOD algorithm is needed. Consider $f_{\hat{\theta}}$ learnt by ERM. If $\mathcal{V}_{\hat{\gamma}|\hat{\theta}}$ is relatively larger than $\tilde{\mathcal{V}}_{\gamma|\theta}$, then ERM fails to avoid varying features. In this case, one should consider an OOD algorithm to achieve better OOD generalization. Otherwise, ERM is enough, and any attempt to achieve better OOD accuracy should start with finding more domains instead of using OOD algorithms. This coincides experiments in Gulrajani & Lopez-Paz (2020) (Section5.2). Our understanding is that domains in $\tilde{\mathbb{S}}$ are similar. Therefore, the difference between shuffle and standard version of the metric reflects how much varying features a learnt model uses. We show how to use the two version of $\mathcal{V}_{\gamma|\theta}$ in Section 5.1.1 and Section 5.2.

### 4.3 INFLUENCE CALCULATION

There is a question surrounding the influence function: how to efficiently calculate and inverse Hessian? Koh & Liang (2017) suggests Conjugate Gradient and Stochastic estimation solve the problem. However, when $\hat{\theta}$ is obtained by running SGD, it could hardly arrive at the global minimum. Although adding a damping term (i.e. let $\hat{H}_{\hat{\theta}} = H_{\hat{\theta}} + \lambda I$) can moderately alleviate the problem by transforming it into a convex situation, under large neural-network with non-linear activation function like ReLU, this method may still work poorly since the damping term in order to satisfy the transform is so large that it will influence the performance significantly. Most importantly, the variation of the eigenvalue of Hessian is huge, making the convergence of influence function calculation quite slow and inaccurate (Basu et al. (2020)).

In our metric, we circumvent the problem by excluding most parameters $\beta$ and directly calculate Hessian of $\gamma$ to get accurate influence function. This modification not only speed up the calculation, but it also coincides our expectation, that an OOD algorithm should learn invariant features *does not mean that* the influence function of *all* parameters should be identical across domains. For example, if $g(\Phi)$ wants to extract the same features in different domains, the influence function should be different on $\Phi(\cdot)$. Therefore, if we use all parameters to calculate the influence, given that $\gamma$ is relatively insignificant in size compared with $\beta$, the information of learnt features provided by $\gamma$ is hard to be captured. On the contrary, only considering the influence of the top model will manifest the influence of different domains in the aspect of *features*, thus enabling us to achieve our goal.

As our experiments show, after this modification, the influence function calculation speed can be 2000 times faster, and the utility (correlation with OOD property) could be even higher. One may not feel surprised given the huge number of parameters in the embedding model $\Phi(\cdot)$. They slow down the calculation and overshadow the top model's influence value.

## 5 EXPERIMENT

In this section, we experimentally show that: (1) A model $f_{\hat{\theta}}$ reaches small $\mathcal{V}_{\hat{\gamma}|\hat{\theta}}$ if it has good OOD property, while a non-OOD model won't. (2) The metric $\mathcal{V}_{\gamma|\theta}$ provides additional information on the stability of a learnt model, which overcomes the weakness of the test accuracy. (3) The comparison of $\mathcal{V}_{\gamma|\beta}$ and $\tilde{\mathcal{V}}_{\gamma|\beta}$ can check whether a better OOD algorithm is needed.

We consider experiments in Bayesian Network, Colored MNIST and VLCS. The synthetic data generated by Bayesian Network includes domain-dependent noise and fake associations between features and response. For Colored MNIST, we already know that the digit is the causal feature and the color is non-causal. The causal relationships help us to determine the worst domain and obtain the OOD accuracy. VLCS is a real dataset, in which we show utility of $\mathcal{V}_{\gamma|\theta}$ step by step. Due to the space limitation, we put the experiments in Bayesian Network to the appendix.

Generally, cross-validation (Gulrajani & Lopez-Paz (2020)) is used to judge a model's OOD property. In the introduction, we have already shown that the leave-one-domain-out cross-validation may fail to discern OOD properties. We also consider another two potential competitors: conditional mutual information and IRM penalty. The comparison between our metric and the two competitors are postponed into Appendix.

### 5.1 COLORED MNIST

Colored MNIST (Arjovsky et al. (2019)) introduces a synthetic binary classification task. The images are colored according to their label, making color a spurious feature in predicting the label. Specifically, for a domain $e$, we assign a preliminary binary label $\tilde{y} = \mathbf{1}_{\text{digits} \leq 4}$ and randomly flip $\tilde{y}$ with $p = 0.25$. Then, we color the image according to $\tilde{y}$ but with a flip rate of $p^e$. Clearly, when $p^e < 0.25$ or $p^e > 0.75$, color is more correlated with $\tilde{y}$ than real digit. Therefore, the oracle OOD model $f_{\text{OOD}}$ will attain accuracy 0.75 in all domains while an ERM model may attain high training accuracy and low OOD property if $p^e$ in training domains is too small or too large. Throughout the Colored MNIST experiments, we use three-layer MLP with ReLU activation and hidden dimension 256. Although our MLP model has relatively many parameters and is non-convex due to the activation layer, due to the technique mentioned in Section 4.3, the in-

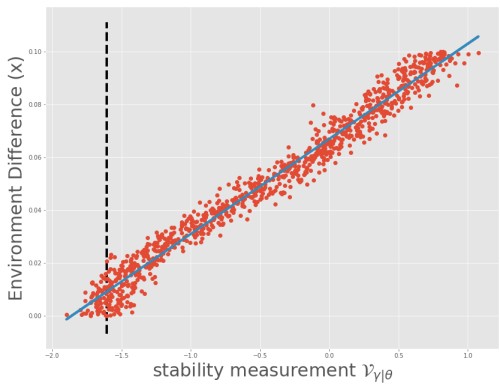

Figure 2: The index $\mathcal{V}_{\gamma|\theta}$ is highly correlated to $x$. The plot contains 501 learnt ERM models with $x = 2 \times 10^{-4} i$, $i = 0, 1, ..., 500$. The dashed line is the baseline value when the difference between domains is eliminated by pooling and redistributing the training data. The blue solid line is the linear regression of $x$ versus $\mathcal{V}_{\gamma|\theta}$.

fluence calculation is still fast and accurate, with directly calculating influence once spends less than 2 seconds.

### 5.1.1 IDENTIFY OOD PROBLEM

In this section, we show that $\mathcal{V}_{\gamma|\theta}$ can discern whether the training domains are sufficiently diverse as mentioned in Section 4.2. Assume $\mathcal{E}_{tr}$ has five training domains with

$$p^e \in \{0.2 - 2x, 0.2 - x, 0.2, 0.2 + x, 0.2 + 2x\},$$

where $x \in [0.0, 0.1]$ is positively related to the diversity among the training domains. If $x$ is zero, all data points are generated from the same domain ($p^e = 0.2$) and so the learning task on $\mathcal{E}_{tr}$ is not an OOD problem. On the contrary, larger $x$ means that the training domains are more diverse. We repeat 501 times to learn the model with ERM. Given the learnt model $f_{\hat{\theta}}$ and the training data, we compute $\mathcal{V}_{\hat{\gamma}|\hat{\theta}}$ and check the correlation between $\mathcal{V}_{\hat{\gamma}|\hat{\theta}}$ and $x$. Figure 2 presents the results. Our index $\mathcal{V}_{\gamma|\theta}$ is highly related to $x$. The Pearson coefficient is $0.9869$, and the Spearman coefficient is $0.9873$. Also, the benchmark of $\mathcal{V}_{\gamma|\theta}$ that learns on the same training domains ($\tilde{\mathbb{S}}$ in 4.2) can be derived from the raw data by pooling and redistributing all data points, and we mark it by the black dashed line. If $\mathcal{V}_{\hat{\gamma}|\hat{\theta}}$ is much higher than the benchmark, indicating that $x$ is not small, an OOD algorithm should be considered if better OOD generalization is demanded. Otherwise, the present algorithm (like ERM) is sufficient. The results coincide our expectation that $\mathcal{V}_{\gamma|\theta}$ can discern whether $P^e$ is different.

### 5.1.2 RELATIONSHIP BETWEEN $\mathcal{V}$ AND OOD ACCURACY

In this section, we use an experiment to support our proposal in Section 4.2. As previously proposed, if a model shows high test accuracy and small $\mathcal{V}_{\gamma|\theta}$ simultaneously, it captures invariant features and avoids varying features, so it deserves to be an OOD model. In this experiment, we consider a model with high test accuracy and show that smaller $\mathcal{V}_{\gamma|\theta}$ generally corresponds to better OOD accuracy, which supports our proposal.

Consider two setups: $p^e \in \{0.0, 0.1\}$ and $p^e \in \{0.1, 0.15, 0.2, 0.25, 0.3\}$. We implement IRM and REx with different penalty (note that ERM is $\lambda = 0$) to check relationship between $\mathcal{V}_{\gamma|\theta}$ and OOD accuracy. For IRM and REx, we run 190 epochs pre-training with $\lambda = 1$ and use early stopping to prevent over-fitting. With this technique, all models successfully achieve good test accuracy (within 0.1 of the oracle accuracy) and meet our requirement. Figure 3 presents the results. We can see that $\mathcal{V}_{\gamma|\theta}$ are highly correlated to OOD accuracy in IRM and REx, with the absolute of Pearson Coefficient never less than $0.8417$. Those models learned with larger $\lambda$ present better OOD property, learning less varying features, and showing smaller $\mathcal{V}_{\gamma|\theta}$. The results are consistent with our proposal, except that when $\lambda$ is large in IRM, $\mathcal{V}_{\gamma|\theta}$ is a little bit unstable. We have carefully

examined the phenomenon and found that it is caused by computational instability when inversing Hessian with eigenvalue quite close to 0. The problem of unstable inversing happens with a low probability and can be addressed by repeating the experiment once or twice.

## 5.2 Domain Generalization: VLCS

In this section, we implement the proposed metric for 4 algorithms: ERM, gDRO, Mixup and IRM on the VLCS image dataset, which is widely used for doamin generalization. We emulate a real scenario with $\mathcal{E}_{all} = \{V, L, C, S\}$ and $\mathcal{E}_{tr} = \mathcal{E}_{all} \backslash \{S\}$. As mentioned in Gulrajani & Lopez-Paz (2020), we use "training-domain validation set" method, i.e. we split a validation set for each $S \in \mathcal{E}_{tr}$ and the test accuracy is defined as the average accuracy amount the three validation sets. Note that, our goal is to use the test accuracy and $\mathcal{V}_{\gamma|\beta}$ to measure the *OOD generalization*, rather than to tune for the SOTA performance on a unseen domain $\{S\}$. Therefore, we do not apply any model selection method and just use the default hyper-parameters in Gulrajani & Lopez-Paz (2020).

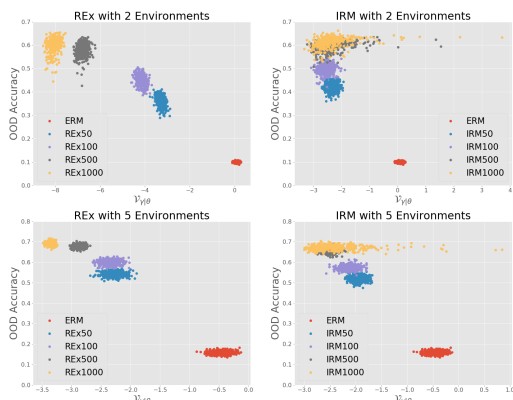

Figure 3: The relationship between $\mathcal{V}_{\gamma|\theta}$ and OOD accuracy in REx (left) and IRM (right) with $\lambda \in \{0, 50, 100, 500, 1000\}$. We train 400 models for each $\lambda$. The OOD accuracy and $\mathcal{V}_{\gamma|\theta}$ enjoy high Pearson coefficient: -0.9745 (up-left), -0.9761 (down-left), -0.8417 (up-right), -0.9476 (down-right). The coefficients are negative because lower $\mathcal{V}_{\gamma|\theta}$ forebodes better OOD property.

### 5.2.1 Step
1: Test accuracy comparison

For each algorithm, we run the naive training process 12 times and show the average of test accuracy of each algorithm in Table 1. Before calculating $\mathcal{V}_{\gamma|\beta}$, the learnt model should at least arrive a good test accuracy. Otherwise, there is no need to discuss its OOD performance since OOD accuracy is smaller than test accuracy. In the table, the test accuracy of ERM, Mixup and gDRO is good, but that of IRM is not. In this case, IRM will be eliminated. If an algorithm fails to reach high test accuracy first, we should first change the hyper-parameters until we observe a relatively high test accuracy.

### 5.2.2 Step 2: shuffle and standard metric comparison

Now we are ready to check whether the learnt models are invariant across $\mathcal{E}_{tr}$. As mentioned in 4.2, the difference of $\mathcal{V}_{\gamma|\beta}$ and $\tilde{\mathcal{V}}_{\gamma|\beta}$ represents whether how much a model is invariant across $\mathcal{E}_{tr}$. We calculate the value and the results are in Figure 4. For ERM and Mixup, the two value is nearly the same. In this case, we expect that ERM and Mixup models are invariant and should have a relatively high OOD accuracy, so no more algorithm is needed. For

Table 1: Step1: Test Accuracy (%)

| Domain | C | L | V | Mean |
|---|---|---|---|---|
| ERM | 99.29 | 73.62 | 77.07 | 83.34 |
| Mixup | 99.32 | 74.36 | 78.84 | 84.17 |
| gDRO | 95.79 | 70.95 | 75.25 | 80.66 |
| IRM | 49.44 | 44.76 | 41.17 | 45.12 |

gDRO, we can clearly see that $\tilde{\mathcal{V}}_{\gamma|\beta}$ is uniformly smaller than $\mathcal{V}_{\gamma|\beta}$. Therefore, gDRO models don't treat different domains equally, and hence we predict that the OOD accuracy will be relatively low. In this case, one who starts with gDRO should turn to other algorithms if a better OOD performance is demanded.

Note that, in the whole process, we know nothing about $\{S\}$, so the OOD accuracy is unseen. However, from the above analysis, we know that (1) in this settings, ERM and Mixup is better than gDRO; (2) one who uses gDRO can turn to other algorithms (like Mixup) for better OOD performance; (3) one who uses ERM should consider collecting more environments if he (she) still

wants to improve OOD performance. So far, we finish the judgement using test accuracy and the proposed metric.

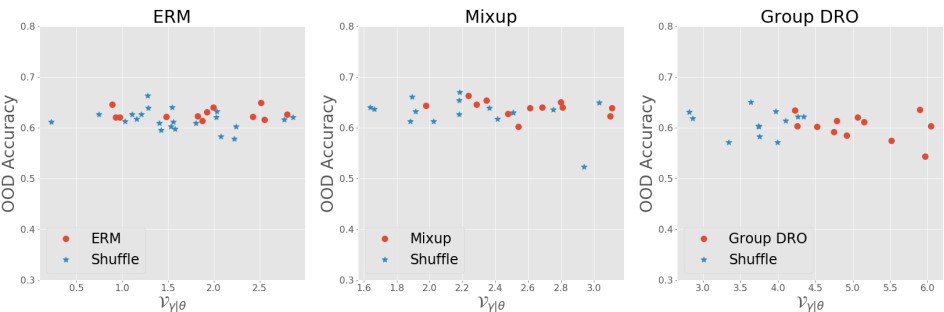

Figure 4: The standard and shuffle version of the metric, i.e. $\mathcal{V}_{\gamma|\beta}$ and $\tilde{\mathcal{V}}_{\gamma|\beta}$ for ERM, Mixup and gDRO. For each algorithm, each version of the metric, we run the experiments more than 12 times in case of statistical error. Similar $\mathcal{V}_{\gamma|\beta}$ and $\tilde{\mathcal{V}}_{\gamma|\beta}$ represents invariance across $\mathcal{E}_{tr}$, which is the case of ERM and Mixup. For gDRO, $\tilde{\mathcal{V}}_{\gamma|\beta}$ is clearly smaller.

### 5.2.3 STEP 3: OOD ACCURACY RESULTS (ORACLE)

In this step, we fortunately obatin $\mathcal{E}_{all}$ and can check whether our judgement is reasonable. Normally, this step will not happen. We now show the OOD accuracy of four algorithms in table 2. Similar to our judgement, ERM and Mixup models achieve a higher OOD accuracy

Table 2: Step3: OOD Accuracy (%)

|  | ERM | Mixup | gDRO | IRM |
|---|---|---|---|---|
| Mean | **62.76** | **63.91** | 60.17 | 31.33 |
| Std | 1.16 | 1.57 | 2.56 | 13.44 |

than gDRO. The performance of IRM (under this hyper-parameters) is lower than test accuracy. During the above process, we can also compare the metric of the model from the same algorithm but with different hyper-parameters (as the same in section 5.1.2). Besides, one may notice that even the highest OOD accuracy is just $63.91\%$. That is to say, to obtain OOD accuracy larger than $70\%$, we should consider collecting more environments. In the appendix A.6, we continue our real scenario to see that, if initially $\mathcal{E}_{tr}$ is more diverse, what will our metric lead us to.

The whole results in VLCS can also be found in the same appendix, and the comparison of the proposed metric with the IRM penalty in formula 4 can be found there too. Besides, we show the comparison with Conditional Mutual Information in the appendix A.5. In summary, we use a realistic task to see how to judge the OOD property of learnt model using the proposed metric and test accuracy. The judgement coincides well with the real OOD performance.

## 6 CONCLUSION

In this paper, we focus on two presently unsolved problems, that how can we discern the OOD property of multiple domains and of learnt models. To this end, we introduce influence function into OOD problem and propose our metric to help solve these issues. Our metric can not only discern whether a multi-domains problem is OOD but can also judge a model's OOD property when combined with test accuracy. To make our calculation more meaningful, accurate and efficient, we modify influence function to domain-level and propose to use only the top model to calculate the influence. Our method is proved in simple cases and it works well in experiments. We sincerely hope that, with the help of this index, our understanding of OOD generalization will become more and more precise and thorough.

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

# A APPENDIX

## A.1 SIMPLE BAYESIAN NETWORK

In this section, we show that the model with better OOD accuracy achieves smaller $\mathcal{V}_{\gamma|\theta}$. We assume the data is generated from the following Bayesian network:

$$\mathbf{x}_1 \leftarrow \mathcal{N}(0, \sigma_e^2), \quad \mathbf{y} \leftarrow \mathbf{x}_1^e \boldsymbol{W}_{1 \rightarrow \mathbf{y}} + \mathcal{N}(0, 1), \quad \mathbf{x}_2 \leftarrow \mathbf{y}^e \boldsymbol{W}_{\mathbf{y} \rightarrow 2} + \mathcal{N}(0, \sigma_e^2). \tag{8}$$

where $\mathbf{x}_1, \mathbf{x}_2 \in \mathbb{R}^5$ are the features, $\mathbf{y} \in \mathbb{R}^5$ is the target vector, $\boldsymbol{W}_{1 \rightarrow \mathbf{y}} \in \mathbb{R}^{5 \times 5}$ and $\boldsymbol{W}_{\mathbf{y} \rightarrow 2} \in \mathbb{R}^{5 \times 5}$ are the underlying parameters that are invariant across domains. The variance of gaussian noise is $\sigma_e^2$ that depends on domain. For simplicity, we denote $e = \sigma_e$ to represent a domain. The goal here is to linearly regress the response y on the input vector $(\boldsymbol{x}_1, \boldsymbol{x}_2)$, i.e. $\hat{\boldsymbol{y}} = \boldsymbol{x}_1 \hat{\boldsymbol{W}}_1 + \boldsymbol{x}_2 \hat{\boldsymbol{W}}_2$. According to the Bayesian network (8), $\mathbf{x}_1$ is the invariant feature, while the correlation between $\mathbf{x}_2$ and $\mathbf{y}$ is spurious and unstable since $e = \sigma_e$ varies across domains. Clearly, the model based only on $\mathbf{x}_1$ is an invariant model. Any invariant estimator should achieve $\hat{\boldsymbol{W}}_1 \approx \boldsymbol{W}_{1 \rightarrow \mathbf{y}}$ and $\hat{\boldsymbol{W}}_2 \approx \mathbf{0}$.

Table 3: Average parameter error $\|\hat{\boldsymbol{W}} - \boldsymbol{W}\|^2$ and the stable measurement $\mathcal{V}_{\gamma|\theta}$ of 500 models from ERM, IRM and REx. Here, "Causal Error" represents $\|\hat{\boldsymbol{W}}_1 - \boldsymbol{W}_{1 \rightarrow \mathbf{y}}\|^2$ and "Non-causal Error" represents $\|\hat{\boldsymbol{W}}_2\|^2$.

| Method | $\mathcal{V}_{\gamma|\theta}$ | Causal Error | Non-causal Error |
|--------|------|--------------|------------------|
| ERM | 15.844 | 0.582 | 0.581 |
| IRM | 5.254 | 0.122 | 0.109 |
| REx | 1.341 | 0.042 | 0.033 |

Now consider five training domains $e \in \mathcal{E}_{tr} = \{0.2, 0.7, 1.2, 1.7, 2.2\}$ , each containing 1000 data points. We estimate three linear models using ERM, IRM and REx respectively and record the parameter error as well as $\mathcal{V}_{\gamma|\theta}$ (note that $\gamma$ is $\theta$ here). Table 3 presents the results among 500 repetitions. As expected, IRM and REx learn more invariant relationships than ERM (smaller causal error) and better avoid non-causal variables ($\hat{\boldsymbol{W}}_2 \approx \mathbf{0}$). Furthermore, the proposed measurement $\mathcal{V}_{\gamma|\theta}$ is highly related to invariance, i.e. model with better OOD property achieves smaller $\mathcal{V}_{\gamma|\theta}$. This results coincides our understanding.

## A.2 PROOF OF AN EXAMPLE

In this section, we use a simple model to illuminate the validity of $\mathcal{V}_{\gamma|\theta}$ proposed in Section 4. Consider a structural equation model (Wright (1921)):

$$\mathrm{x}_1 \sim P_x^e, \quad \mathrm{y} \leftarrow \mathrm{x}_1 + \mathcal{N}(0,1), \quad \mathrm{x}_2 \leftarrow \mathrm{y} + \mathcal{N}(0,\sigma_e^2)$$

where $P_x^e$ is a distribution with a finite second-order moment, i.e. $\mathbb{E}\mathrm{x}_1^2 < +\infty$, and $\sigma_e^2$ is the variance of the noise term in $\mathrm{x}_2$. Both $P_x^e$ and $\sigma_e^2$ vary across domains. For simplicity, we assume there are infinite training data points collected from two training domains $\mathcal{E}_{tr} = \{(P_x^1, \sigma_1^2), (P_x^2, \sigma_2^2)\}$. Our goal is to predict y from $\mathbf{x} := (\mathrm{x}_1, \mathrm{x}_2)^\top$ using a least-squares predictor $\hat{y} = \mathbf{x}^\top \hat{\boldsymbol{\beta}} := \mathrm{x}_1 \hat{\beta}_1 + \mathrm{x}_2 \hat{\beta}_2$. Here we consider two algorithms: ERM and IRM with $\lambda \to +\infty$. According to Arjovsky et al. (2019), using IRM we obtain $\boldsymbol{\beta}_{\mathrm{IRM}} \to (1,0)^\top$.

Intuitively, ERM will exploit both $\mathrm{x}_1$ and $\mathrm{x}_2$, thus achieving a better regression model. However, since relationship between y and $\mathrm{x}_2$ varies across domains, our index will be huge in such condition. Conversely, $\beta_{\mathrm{IRM}}$ only uses invariant features $\mathrm{x}_1$, thus $\mathcal{V}_{\gamma|\theta} \to -\infty$. Note that we do not have an embedding model here, so $\mathcal{V}_{\gamma|\theta} = \mathcal{V}_{\boldsymbol{\beta}}$.

**ERM** we denote

$$\ell(\boldsymbol{\beta}) = \frac{1}{|\mathcal{E}_{tr}|} \sum_{e \in \mathcal{E}_{tr}} \ell_e(\boldsymbol{\beta}) \quad \text{with} \quad \ell_e(\boldsymbol{\beta}) = \mathbb{E}_e (\mathrm{y} - \mathbf{x}\boldsymbol{\beta})^2.$$

Note that in $\mathbb{E}_e$, $\mathrm{x}_1$ is sample from $P_x^e$. We then have

$$\frac{\partial \ell(\boldsymbol{\beta})}{\boldsymbol{\beta}} = -\sum_{e \in \mathcal{E}_{tr}} \mathbb{E}_e[\mathbf{x}(\mathrm{y} - \mathbf{x}^\top \boldsymbol{\beta})] = -\frac{2}{|\mathcal{E}_{tr}|} \sum_{e \in \mathcal{E}_{tr}} \left( \begin{array}{c} \mathbb{E}_e[\mathrm{x}_1(\mathrm{y} - \mathbf{x}^\top \boldsymbol{\beta})] \\ \mathbb{E}_e[\mathrm{x}_2(\mathrm{y} - \mathbf{x}^\top \boldsymbol{\beta})] \end{array} \right)$$

To proceed further, we denote

$$\bar{d} = \frac{1}{|\mathcal{E}_{tr}|} \sum_{e \in \mathcal{E}_{tr}} \mathbb{E}_e \mathrm{x}_1^2, \quad s = \sum_{e \in \mathcal{E}_{tr}} \sigma_e^2 = \sigma_1^2 + \sigma_2^2.$$

By solving the following equations:

$$\frac{1}{|\mathcal{E}_{tr}|} \sum_{e \in \mathcal{E}_{tr}} \mathbb{E}_e[\mathrm{x}_1(\mathrm{y} - \mathbf{x}^\top \boldsymbol{\beta})] = \bar{d}(1 - \beta_1 - \beta_2) = 0$$

and

$$\frac{1}{|\mathcal{E}_{tr}|} \sum_{e \in \mathcal{E}_{tr}} \mathbb{E}_e[\mathrm{x}_2(\mathrm{y} - \mathbf{x}^\top \boldsymbol{\beta})] = (\bar{d} + 1)(1 - \beta_1 - \beta_2) + \beta_1 - \frac{s}{|\mathcal{E}_{tr}|} \beta_2 = 0$$

we have $\hat{\boldsymbol{\beta}} = (\hat{\beta}_1, \hat{\beta}_2)^\top$ with

$$\hat{\beta}_1 = \frac{s}{s+2}, \quad \hat{\beta}_2 = \frac{2}{s+2}.$$

Now we calculate our index. It is easy to see that

$$\frac{\partial \ell_e(\boldsymbol{\beta})}{\beta_1} = -2\mathbb{E}_e[\mathrm{x}_1(\mathrm{y} - \mathbf{x}^\top \boldsymbol{\beta})] = -2\mathbb{E}_e \mathrm{x}_1^2(1 - \beta_1 - \beta_2)$$

$$\frac{\partial \ell_e(\boldsymbol{\beta})}{\beta_2} = -2\mathbb{E}_e[\mathrm{x}_2(\mathrm{y} - \mathbf{x}^\top \boldsymbol{\beta})] = -2[(\mathbb{E}_e \mathrm{x}_1^2 + 1)(1 - \beta_1 - \beta_2) + \beta_1 - \sigma_e^2 \beta_2].$$

Therefore,

$$\nabla \ell_1(\boldsymbol{\beta}) - \nabla \ell_2(\boldsymbol{\beta}) = \left( \begin{array}{c} 0 \\ 2\beta_2(\sigma_1^2 - \sigma_2^2) \end{array} \right) \quad \text{and} \quad \nabla \ell_1(\hat{\boldsymbol{\beta}}) - \nabla \ell_1(\hat{\boldsymbol{\beta}}) = \left( \begin{array}{c} 0 \\ \frac{4(\sigma_1^2 - \sigma_2^2)}{s+2} \end{array} \right) \quad (9)$$

On the other hand, calculate the hessian and we have

$$\boldsymbol{H}_{\mathrm{ERM}} = \left( \begin{array}{cc} 2\bar{d} & 2\bar{d} \\ 2\bar{d} & 2\bar{d} + s + 2 \end{array} \right) \quad \text{and} \quad \boldsymbol{H}^{-1} = \frac{1}{2\bar{d}(s+2)} \left( \begin{array}{cc} 2\bar{d} + s + 2 & -2\bar{d} \\ -2\bar{d} & 2\bar{d} \end{array} \right).$$

Then we have (note that $\mathcal{IF}(\hat{\boldsymbol{\beta}}, \mathbb{S}^e) = \boldsymbol{H}^{-1} \nabla \ell_e(\hat{\boldsymbol{\beta}})$)

$$
\begin{aligned}
\mathcal{V}_{\hat{\boldsymbol{\beta}}} &= \ln(\|\text{Cov}_{e \in \mathcal{E}}(\mathcal{IF}(\hat{\boldsymbol{\beta}}, \mathbb{S}^e))\|_2) \\
&= \ln(\frac{1}{4}\|(\mathcal{IF}_1 - \mathcal{IF}_2)(\mathcal{IF}_1 - \mathcal{IF}_2)^\top\|_2) \\
&= \ln(\frac{1}{4}\|\mathcal{IF}_1 - \mathcal{IF}_2\|^2) \\
&= 2\ln(\frac{1}{2}\|\boldsymbol{H}^{-1}(\nabla \ell_1(\hat{\boldsymbol{\beta}}) - \nabla \ell_2(\hat{\boldsymbol{\beta}}))\|) \\
&= 2\ln(\frac{1}{4\bar{d}(s+2)}\|\begin{pmatrix} 2\bar{d} + s + 2 & -2\bar{d} \\ -2\bar{d} & 2\bar{d} \end{pmatrix}\begin{pmatrix} 0 \\ \frac{4(\sigma_1^2 - \sigma_2^2)}{s+2} \end{pmatrix}\|) \\
&= 2\ln(\frac{2\sqrt{2}|\sigma_1^2 - \sigma_2^2|}{(s+2)^2})
\end{aligned}
$$

where the third equation holds because the rank of matrix is 1. Clearly, when $|\sigma_1^2 - \sigma_2^2| \to 0$ (means two domains become identical), our index $\mathcal{V}_{\boldsymbol{\beta}} \to -\infty$. Otherwise, given $\sigma_1 \neq \sigma_2$, we have $\mathcal{V}_{\boldsymbol{\beta}} > -\infty$, showing that ERM captures varied features.

**IRM** We now turn to IRM model and show that $\mathcal{V}_{\boldsymbol{\beta}} \to -\infty$ when $\lambda \to +\infty$, thus proving IRM learnt model $\hat{\boldsymbol{\beta}}_{\text{IRM}}$ does achieve smaller $\mathcal{V}_{\boldsymbol{\beta}}$ compared with $\hat{\boldsymbol{\beta}}$ in ERM.

Under IRM model, assuming the tuning parameter is $\lambda$, we have

$$
\mathcal{L}(\boldsymbol{\beta}) = \frac{1}{|\mathcal{E}_{tr}|}\sum_{e \in \mathcal{E}_{tr}} \mathbb{E}_e[(\mathbf{y} - \mathbf{x}^\top\boldsymbol{\beta})^2] + 4\lambda\|\mathbb{E}_e[\mathbf{x}^\top\boldsymbol{\beta}(\mathbf{y} - \mathbf{x}^\top\boldsymbol{\beta})]\|^2.
$$

Then we have the gradient with respect $\boldsymbol{\beta}$:

$$
\nabla\mathcal{L}(\boldsymbol{\beta}) = \frac{1}{|\mathcal{E}_{tr}|}\sum_{e \in \mathcal{E}_{tr}}\left(-2\mathbb{E}_e[\mathbf{x}(\mathbf{y} - \mathbf{x}^\top\boldsymbol{\beta})] + 8\lambda\mathbb{E}_e[\mathbf{x}^\top\boldsymbol{\beta}(\mathbf{y} - \mathbf{x}^\top\boldsymbol{\beta})]\mathbb{E}_e[\mathbf{x}(\mathbf{y} - 2\mathbf{x}^\top\boldsymbol{\beta})]\right),
$$

and the Hessian matrix

$$
\boldsymbol{H} = \boldsymbol{H}_{\text{ERM}} + \frac{8\lambda}{|\mathcal{E}|}\sum_{e \in \mathcal{E}}\left(\mathbb{E}_e[\mathbf{x}(\mathbf{y} - 2\mathbf{x}^\top\boldsymbol{\beta})]\mathbb{E}_e[\mathbf{x}(\mathbf{y} - 2\mathbf{x}^\top\boldsymbol{\beta})]^\top - 2\mathbb{E}_e[\mathbf{x}^\top\boldsymbol{\beta}(\mathbf{y} - \mathbf{x}^\top\boldsymbol{\beta})]\mathbb{E}_e[\mathbf{x}\mathbf{x}^\top]\right).
$$

$$(10)$$

Denote $\boldsymbol{\beta}_\lambda$ the solution of IRM algorithm on $\mathcal{E}_{tr}$ when penalty is $\lambda$. From Arjovsky et al. (2019) we know $\boldsymbol{\beta}_\lambda \to \boldsymbol{\beta}_{\text{IRM}} := (1, 0)^\top$. To show $\lim_{\lambda \to +\infty} \mathcal{V}_{\boldsymbol{\beta}_\lambda} = -\infty$, we only need to show that

$$
\lim_{\lambda \to +\infty} \boldsymbol{H}^{-1}(\nabla \ell_1(\boldsymbol{\beta}_\lambda) - \nabla \ell_2(\boldsymbol{\beta}_\lambda)) = \mathbf{0}
$$

We prove this by showing that

$$
\lim_{\lambda \to +\infty} \boldsymbol{H}^{-1}(\lambda) = \mathbf{0} \quad \text{and} \quad \lim_{\lambda \to +\infty} \nabla \ell_1(\boldsymbol{\beta}_\lambda) - \nabla \ell_2(\boldsymbol{\beta}_\lambda) = \mathbf{0} \tag{11}
$$

simultaneously. We add $(\lambda)$ after $\boldsymbol{H}^{-1}$ to show that $\boldsymbol{H}^{-1}$ is a continuous function of $\lambda$. Rewrite $\boldsymbol{H}$ in formula 10 as

$$
\boldsymbol{H}(\lambda, \boldsymbol{\beta}_\lambda) = \boldsymbol{H}_{\text{ERM}} + \lambda F(\boldsymbol{\beta}_\lambda)
$$

where

$$
F(\boldsymbol{\beta}) = \frac{8}{|\mathcal{E}_{tr}|}\sum_{e \in \mathcal{E}_{tr}}\left(\mathbb{E}_e[\mathbf{x}(\mathbf{y} - 2\mathbf{x}^\top\boldsymbol{\beta})]\mathbb{E}_e[\mathbf{x}(\mathbf{y} - 2\mathbf{x}^\top\boldsymbol{\beta})]^\top - 2\mathbb{E}_e[\mathbf{x}^\top\boldsymbol{\beta}(\mathbf{y} - \mathbf{x}^\top\boldsymbol{\beta})]\mathbb{E}_e[\mathbf{x}\mathbf{x}^\top]\right)
$$

$$
\lim_{\boldsymbol{\beta}_\lambda \to \boldsymbol{\beta}_{\text{IRM}}} F(\boldsymbol{\beta}_\lambda) = \frac{4}{|\mathcal{E}_{tr}|}\sum_{e \in \mathcal{E}_{tr}}\begin{pmatrix} -\mathbb{E}_e\mathbf{x}_1^2 \\ 1 - \mathbb{E}_e\mathbf{x}_1^2 \end{pmatrix}\begin{pmatrix} -\mathbb{E}_e\mathbf{x}_1^2 & 1 - \mathbb{E}_e\mathbf{x}_1^2 \end{pmatrix} = F(\boldsymbol{\beta}_{\text{IRM}}) \text{ exists.}
$$

Obviously, $F(\boldsymbol{\beta}_{\text{IRM}})$ is positive definite. Therefore, we have

$$
\begin{aligned}
\lim_{\lambda \to +\infty} \boldsymbol{H}(\lambda, \boldsymbol{\beta}_\lambda)^{-1} &= \lim_{\lambda \to +\infty} \lim_{\boldsymbol{\beta}_\lambda \to \boldsymbol{\beta}_{\text{IRM}}} [\boldsymbol{H}_{\text{ERM}} + \lambda F(\boldsymbol{\beta}_\lambda)]^{-1} \\
&= \lim_{\lambda \to +\infty} [\boldsymbol{H}_{\text{ERM}} + \lambda F(\boldsymbol{\beta}_{\text{IRM}})]^{-1} \\
&= \boldsymbol{0}
\end{aligned}
$$

The first equation holds because $\lim_{\lambda \to +\infty} F(\boldsymbol{\beta}_\lambda) = F(\boldsymbol{\beta}_{\text{IRM}})$ has the limit and is not $\boldsymbol{0}$, and the last equation holds because the eigenvalue of $\boldsymbol{H}$ goes to $+\infty$ when $\lambda \to +\infty$.

Now consider $\nabla \ell_1(\boldsymbol{\beta}_\lambda) - \nabla \ell_2(\boldsymbol{\beta}_\lambda)$. According to formula 9, we have

$$
\begin{aligned}
\lim_{\lambda \to +\infty} \nabla \ell_1(\boldsymbol{\beta}_\lambda) - \nabla \ell_2(\boldsymbol{\beta}_\lambda) &= \lim_{\boldsymbol{\beta}_\lambda \to \boldsymbol{\beta}_{\text{IRM}}} \nabla \ell_1(\boldsymbol{\beta}_\lambda) - \nabla \ell_2(\boldsymbol{\beta}_\lambda) \\
&= \nabla \ell_1(\boldsymbol{\beta}_{\text{IRM}}) - \nabla \ell_2(\boldsymbol{\beta}_{\text{IRM}}) \\
&= \begin{pmatrix} 0 \\ 2\beta_2(\sigma_1^2 - \sigma_2^2) \end{pmatrix} \\
&= \boldsymbol{0}
\end{aligned}
$$

Hence we finish proof of formula 11 and show that $\mathcal{V}_{\boldsymbol{\beta}} \to -\infty$ in IRM.

## A.3 Formula (5)

This section shows the derivation of the expression (5). Recall that the training dataset $\mathbb{S} = \{\mathbb{S}^1, ..., \mathbb{S}^m\}$ and the objective function

$$
\mathcal{L}(f, \mathbb{S}) = \ell(f, \mathbb{S}) + \lambda R(f, \mathbb{S}),
$$

where the second term on the right hand side is the regularization. As to ERM, the regularization term is zero. With the feature extractor ($\boldsymbol{\beta}$) fixed, we upweight a domain $\mathbb{S}^e$. The new objective function is

$$
\mathcal{L}_+(\boldsymbol{\theta}, \mathbb{S}, \delta) = \mathcal{L}(\boldsymbol{\theta}, \mathbb{S}) + \delta \cdot \ell(\boldsymbol{\theta}, \mathbb{S}^e)
$$

Notice that when upweight an domain, we only upweight the empirical loss on the corresponding domain. Further, we denote $\hat{\boldsymbol{\gamma}}, \hat{\boldsymbol{\gamma}}_+$ as the optimal solutions before and after upweighting a domain. It is easy to see that $\|\hat{\boldsymbol{\gamma}}_+ - \hat{\boldsymbol{\gamma}}\| \to 0$ when $\delta \to 0$. Following the derivation in Koh & Liang (2017), according to the first-order Taylor expansion of $\nabla_{\boldsymbol{\gamma}} \mathcal{L}_+(\boldsymbol{\theta}, \mathbb{S}, \delta)$ with respect to $\boldsymbol{\gamma}$ on $\hat{\boldsymbol{\gamma}}$,

$$
\begin{aligned}
\boldsymbol{0} &= \nabla_{\boldsymbol{\gamma}}[\mathcal{L}(\hat{\boldsymbol{\theta}}_+, \mathbb{S}) + \delta \ell(\hat{\boldsymbol{\theta}}_+, \mathbb{S}^e)] \\
&= \nabla_{\boldsymbol{\gamma}}(\mathcal{L}(\hat{\boldsymbol{\theta}}, \mathbb{S}) + \delta \ell(\hat{\boldsymbol{\theta}}, \mathbb{S}^e)) + \nabla_{\boldsymbol{\gamma}}^2[\mathcal{L}(\hat{\boldsymbol{\theta}}, \mathbb{S}) + \delta \ell(\hat{\boldsymbol{\theta}}, \mathbb{S}^e)](\hat{\boldsymbol{\gamma}}_+ - \hat{\boldsymbol{\gamma}}) + o(\|\hat{\boldsymbol{\gamma}}_+ - \hat{\boldsymbol{\gamma}}\|) \\
&= \delta \nabla_{\boldsymbol{\gamma}} \ell(\hat{\boldsymbol{\theta}}, \mathbb{S}^e) + \nabla_{\boldsymbol{\gamma}}^2[\mathcal{L}(\hat{\boldsymbol{\theta}}, \mathbb{S}) + \delta \ell(\hat{\boldsymbol{\theta}}, \mathbb{S}^e)](\hat{\boldsymbol{\gamma}}_+ - \hat{\boldsymbol{\gamma}}) + o(\|\hat{\boldsymbol{\gamma}}_+ - \hat{\boldsymbol{\gamma}}\|)
\end{aligned}
$$

Assume that $\nabla^2[\mathcal{L}(\hat{\boldsymbol{\theta}}, \mathbb{S}) + \delta \ell(\hat{\boldsymbol{\theta}}, \mathbb{S}^e)]$ is invertible, we have

$$
\begin{aligned}
\frac{\hat{\boldsymbol{\gamma}}_+ - \hat{\boldsymbol{\gamma}}}{\delta} &= [\nabla_{\boldsymbol{\gamma}}^2 \mathcal{L}(\hat{\boldsymbol{\theta}}, \mathbb{S}) + \delta \ell(\hat{\boldsymbol{\theta}}, \mathbb{S}^e)]^{-1} \nabla_{\boldsymbol{\gamma}} \ell(\hat{\boldsymbol{\theta}}, \mathbb{S}^e) + o(\|\frac{\hat{\boldsymbol{\gamma}}_+ - \hat{\boldsymbol{\gamma}}}{\delta}\|) \\
\lim_{\delta \to 0} \frac{\hat{\boldsymbol{\gamma}}_+ - \hat{\boldsymbol{\gamma}}}{\delta} &= [\nabla_{\boldsymbol{\gamma}}^2 \mathcal{L}(\hat{\boldsymbol{\theta}}, \mathbb{S})]^{-1} \nabla_{\boldsymbol{\gamma}} \ell(\hat{\boldsymbol{\theta}}, \mathbb{S}^e)
\end{aligned}
$$

Note that this derivation is not fully rigorous. Please refer to Van der Vaart (2000) for more rigorous discussions about influence function.

The reason that $\boldsymbol{\beta}$ should be fixed is as follows. First, if $\boldsymbol{\beta}$ can be varied, then the change of $\boldsymbol{\theta}$ will become:

$$
\begin{pmatrix} H_{\gamma\gamma} & H_{\gamma\beta} \\ H_{\beta\gamma} & H_{\beta\beta} \end{pmatrix}^{-1} \begin{pmatrix} \nabla_{\boldsymbol{\gamma}} l(\hat{\boldsymbol{\theta}}, \mathbb{S}^e) \\ \nabla_{\boldsymbol{\beta}} l(\hat{\boldsymbol{\theta}}, \mathbb{S}^e) \end{pmatrix}.
$$

Therefore, the computational cost is similar to calculate and inverse the whole hessian matrix. Most importantly, without fixing $\boldsymbol{\beta}$, the change of $\boldsymbol{\gamma}$ is somehow useless. Say when upweighting $\mathbb{S}^e$, the use of a feature decreases. It's possible, however, that the parameter in $\boldsymbol{\gamma}$ corresponding to the feature increases while $\boldsymbol{\beta}$ decreases a larger scale. In this case, the use of the feature decreases but $\boldsymbol{\gamma}$ increases. Without fixing $\boldsymbol{\beta}$, the change of $\boldsymbol{\gamma}$ calculated by influence function may provide no information about the use a feature. Therefore, we argue that, fixing $\boldsymbol{\beta}$ is a "double-win" choice.

## A.4 ACCURACY IS NOT ENOUGH

In Introduction, we have given an example where test accuracy misleads us. In this section, we will first supplement some examples where test accuracy not only misjudge different algorithms, but it also misjudges the OOD property of models learnt with different penalty within the same algorithm. After that, we will show the universality of these problems and why test accuracy fails.

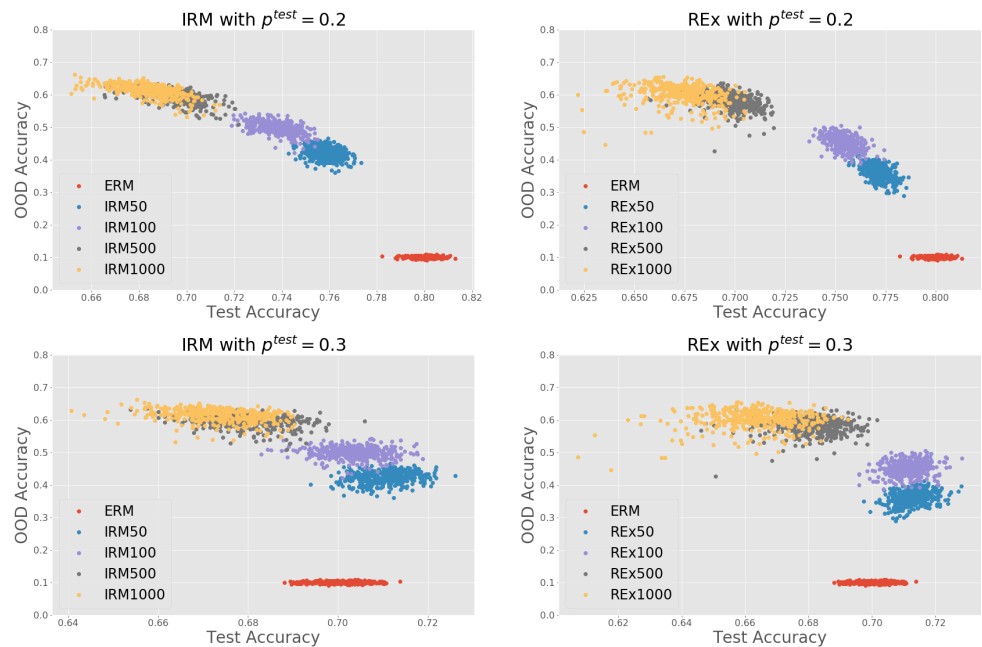

Figure 5: Experiments in Colored MNIST to show test accuracy (x-axis) cannot be used to judge model learnt with different penalty. Consider two test domains with $p^{\text{test}} = 0.2$ (up penals) and $p^{\text{test}} = 0.3$ (down penals). For each $\lambda$, we run IRM and REx 500 times. We can see that when $\lambda$ increases from $\lambda = 0$ to $\lambda = 1000$, the OOD accuracy also increases, but test accuracy does not. When $p^{test} = 0.3$, their relationship becomes more perplexed.

Consider two training domains

$$p^e \in \{0.0, 0.1\},$$

and a test domain with flip rate denoted by $p^{\text{test}}$. We implement IRM and REx with penalty $\lambda \in \{0, 50, 100, 500, 1000\}$ to check the relationship between test accuracy and OOD accuracy. The training process is identical to the experiment in section 5.1.2. As results showed in Figure 5, when OOD property of model gradually improves (caused by gradually increasing $\lambda$), its relationship with test accuracy is either completely (when $p^{\text{test}}$ is 0.2) or partly (when $p^{\text{test}}$ is 0.3) negatively correlated. This phenomenon reveals the weakness of test accuracy. If one wants to select a $\lambda$ when $p^{\text{test}}$ is 0.3, judged by test accuracy, $\lambda = 50$ may be the best choice, no matter in IRM or REx. However, the model learnt with $\lambda = 50$ has OOD accuracy even *less than a random guess model*.

Whether test accuracy is positively, negatively correlated or irrelevant to model's OOD property mainly depends on the "distance" between test domain and the "worst" domain for the model. If test accuracy happens to be the lowest among all the domains, we directly have OOD accuracy equals to test accuracy. In practice, however, their distance may be huge, and this is precisely the difficulty of OOD generalization. For example, we are accessible to images of cows in grasslands, woods and forests, but cows in desert are rare. At this point, the "worst" domain is certainly far from what we can get. If we expect a model to capture the real feature of cows, the model should avoid any usage of background color. However, a model based on color will perform consistently well (better than any OOD model) no matter in grasslands, woods and forests since all of the available domains are green background in general. In Colored MNIST, test accuracy fails in the same way.

Such situations are quite common. Generally, within domains we have, there may be some features that are strongly correlated to the prediction but are slightly varied across domains. These features are spurious, given that their relationship with prediction is significantly disparate in other domains to which we want to generalize. However, using these features in prediction will easily achieve high test accuracy. Consequently, it will be extremely risky to judge models merely by test accuracy.

### A.5 CONDITIONAL MUTUAL INFORMATION

A possible alternative of $\mathcal{V}_{\gamma|\theta}$ may be Conditional Mutual Information (CMI). For three continuous random variables $X, Y, Z$, the CMI is defined as

$$I(X;Y|Z) = \int \int \int p(x,y,z) \log \frac{p(x,y,z)}{p(x,z)p(y|z)} dxdydz \tag{12}$$

where $p(\cdot)$ is the probability density function. Consider $I(e; y|\Phi(x))$ or $I(e; y|\hat{y})$, i.e. the mutual information of $e$ and true label $y$, given the features or the prediction $\hat{y}$ of $x$. The insight is that, if the model is invariant across different domains, then little information about $e$ should be contained in $y$ given $\Phi(x)$. Otherwise, if the prediction $\hat{y}$ is highly correlated to $e$, then the mutual information will be high.

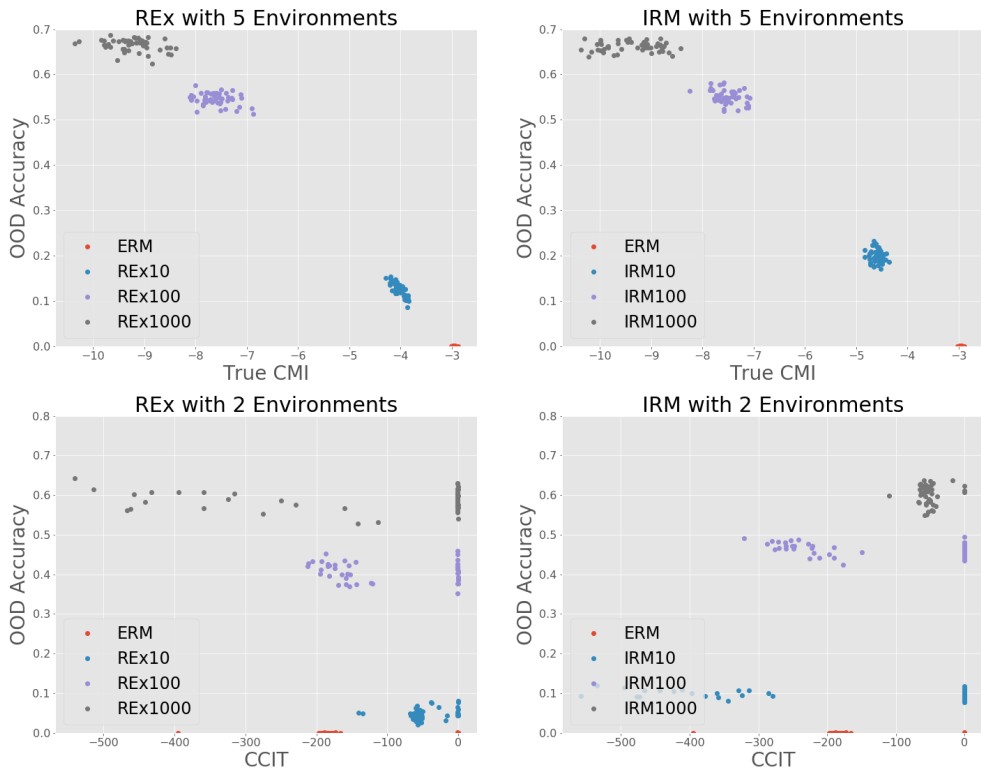

Figure 6: Experiments of the relationship between OOD accuracy and CMI (true or estimated using the method in Sen et al. (2017)). Models are trained by REx (left) and IRM (right) with $\lambda \in \{0, 10, 100, 1000\}$. We train 50 models for each $\lambda$ and calculate the true CMI $I(e; y|\hat{y})$ or CCIT value. As analyzed in the appendix A.5, true CMI enjoys a highly correlated relationship to OOD accuracy, with Pearson Coefficient $-0.9923$ (left) and $-0.9858$ (right). However, the estimated value shows a completely different picture, with Pearson Coefficient $-0.0768$ (left) and $-0.1193$ (right).

This metric seems to be promising. However, the numerical estimation of CMI remains a challenge. To this end, previous works have done a lot to solve this problem, including CCMI proposed in Mukherjee et al. (2020) and CCIT proposed in Sen et al. (2017). In this part, we will first calculate

true $I(e; y|\hat{y})$ in a simple Colored MNIST experiment to show that if there is no estimation problem, CMI could be a potential metric to judge the OOD property of the learnt model, at least in a simple, discrete task. We then run the code provided by Sen et al. (2017) (`https://github.com/rajatsen91/CCIT`) to show that even in this simple task, the estimation of CMI may severely influence its performance.

Specifically, the experimental setting is similar to that in subsection 5.1.2, with two OOD algorithm and number of training domains in $\{2, 5\}$. For each algorithm, we consider the penalty weight $\lambda \in \{0, 10, 100, 1000\}$, run the algorithm 50 times, and record their OOD accuracy as well as true CMI value or CCIT value. The results are shown in Figure 6. We can see that in the case when true CMI can be easily calculated, especially in the case when the number of domains is small and the task is discrete (not continuous), CMI is highly correlated to OOD accuracy. However, in a regression task or in a task when directly calculating the value of CMI becomes impractical, the estimation process may severely destroy the correlation, and may also result in an inverse correlation. Therefore, we summarized the estimation of CMI has limited its utility. We leave the fine-grained analysis of the relationship between CMI, estimated CMI and OOD property to future works.

## A.6    RESULTS ON VLCS

### A.6.1    CONTINUED SCENARIO

This is a continuation of section 5.2. Say in this task, $\mathcal{E}_{all}$ remains the four domains but $_cE_{tr} = \{L, S, V\}$ (empirically we find it more diverse). Similarly, we start with test accuracy shown in table 4. In this step, the situation is the same, i.e. IRM should be eliminated until proper hyper-parameters are found. In step 2, we show the comparison between $\mathcal{V}_{\gamma|\beta}$ and $\tilde{\mathcal{V}}_{\gamma|\beta}$ of three algorithms in Figure 7. As we can see, this time the two value are similar for

Table 4: Domain $C$ out: Test Accuracy (%)

| Domain | L | S | V | Mean |
|--------|-------|-------|-------|-------|
| ERM | 73.43 | 73.87 | 79.15 | 75.48 |
| Mixup | 73.74 | 74.54 | 78.65 | 75.64 |
| gDRO | 71.40 | 71.95 | 77.19 | 73.51 |
| IRM | 49.61 | 38.64 | 45.35 | 44.53 |

all three algorithms, including gDRO. This is different from the case when $S$ is unseen. In this case, we predict that all of the three algorithms should achieve high OOD accuracy. In fact, if we act as the oracle and calculate their OOD performance, we will find that our judgement is close to the reality: ERM, Mixup and gDRO achieve OOD accuracy from $70.55\%$ to $72.87\%$. According to the confidence interval, they difference are not satistically significant. As for IRM, the OOD accuracy is $38.64\%$. One who use ERM, Mixup or gDRO should be satisfied for the performance, since higher demand is somehow impractical!

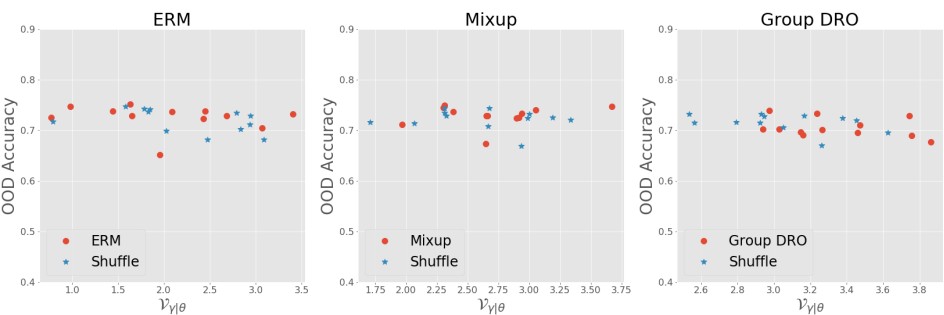

Figure 7: The standard and shuffle version of the metric, i.e. $\mathcal{V}_{\gamma|\beta}$ and $\tilde{\mathcal{V}}_{\gamma|\beta}$ for ERM, Mixup and gDRO. This time, all three algorithms show similar $\mathcal{V}_{\gamma|\beta}$ and $\tilde{\mathcal{V}}_{\gamma|\beta}$.

### A.6.2    FULL RESULTS AND COMPARISON WITH IRM PENALTY

As mentioned in Section 5.2, we consider ERM, gDRO, Mixup and IRM on VLCS image dataset. We report the full results here, and compare the performance of out metric $\mathcal{V}_{\gamma|\theta}$ with IRM penalty in

formula 4. Thorough the whole experiments, $\mathcal{E}_{all} = \{V, L, C, S\}$. We construct four experimental settings. In each setting, one domain is removed and the rest consists of $\mathcal{E}_{tr}$. For each domain in $\mathcal{E}_{tr}$, we split a validation set, and test accuracy is the average accuracy amount validation sets. The results are shown in Table 5. First, our results coincide with Gulrajani & Lopez-Paz (2020) that ERM nearly outperforms any algorithms. We can see that the OOD accuracy of ERM is either the highest or only slightly lower than Mixup. Meanwhile, it has a relatively small $\mathcal{V}_{\gamma|\theta}$. Second, higher OOD accuracy corresponds to lower $\mathcal{V}_{\gamma|\theta}$. In addition, we notice that IRM has a relatively low test accuracy and OOD accuracy. We explain the phenomenon by an improper hyper-parameters in IRM, although we didn't change the default hyper-parameters in the code of Gulrajani & Lopez-Paz (2020) (`https://github.com/facebookresearch/DomainBed`). No matter what, this phenomenon provides a good example in which we can compare our metric with IRM penalty and discuss their advantages and disadvantages.

Table 5: Experiments in VLCS with 4 algorithms. OOD accuracy means the min accuracy in $\mathcal{E}_{all}$. We use training-domain validation method mentioned in Gulrajani & Lopez-Paz (2020), so test accuracy is the average accuracy of three split validation set. "Domain" means which domain is excluded, i.e. which domain is in $\mathcal{E}_{all} \backslash \mathcal{E}_{tr}$. In each setting, we run each algorithm 12 times and report the mean and (std). Note that in a real implementation, IRM penalty can be negative.

| OOD accuracy (%) | | | | | Test accuracy (%) | | | | |
|---|---|---|---|---|---|---|---|---|---|
| Domain | C | L | S | V | Domain | C | L | S | V |
| ERM | 72.54 | 61.48 | 62.76 | 65.59 | ERM | 75.48 | 84.55 | 83.33 | 81.49 |
| | (2.62) | (2.31) | (1.16) | (2.27) | | (3.37) | (10.61) | (11.64) | (12.83) |
| Mixup | 72.87 | 62.10 | 63.91 | 63.81 | Mixup | 75.65 | 84.92 | 84.17 | 81.02 |
| | (2.04) | (3.10) | (1.57) | (3.64) | | (2.80) | (10.54) | (11.07) | (13.52) |
| gDRO | 70.55 | 61.64 | 60.17 | 62.35 | gDRO | 73.51 | 82.82 | 80.66 | 80.03 |
| | (1.91) | (3.92) | (2.56) | (2.11) | | (3.27) | (9.41) | (11.17) | (11.50) |
| IRM | 38.64 | 38.84 | 31.33 | 39.50 | IRM | 44.53 | 48.83 | 45.13 | 51.94 |
| | (0.54) | (0.31) | (13.44) | (2.35) | | (4.94) | (10.53) | (16.53) | (11.66) |
| $\mathcal{V}_{\gamma|\theta}$ (our metric) | | | | | IRM penalty (e-4) | | | | |
| Domain | C | L | S | V | Domain | C | L | S | V |
| ERM | 2.0468 | 1.9084 | 1.8476 | 1.9811 | ERM | 1.78 | 1.48 | 1.43 | 1.63 |
| | (0.3474) | (0.3231) | (0.2887) | (0.3955) | | (1.88) | (1.03) | (0.75) | (2.04) |
| Mixup | 2.6996 | 2.4417 | 2.5810 | 2.8780 | Mixup | 75.4 | 57.7 | 65.5 | 48.3 |
| | (0.1926) | (0.2003) | (0.1492) | (0.2304) | | (32.6) | (26.4) | (37.2) | (30.6) |
| gDRO | 3.3371 | 4.8520 | 5.0915 | 5.1675 | gDRO | 9.42 | 2.13 | 2.6 | 1.94 |
| | (0.1385) | (0.2515) | (0.278) | (0.3507) | | (10.1) | (3.41) | (2.46) | (4.37) |
| IRM | 8.1820 | 6.8329 | 7.6234 | 8.1288 | IRM | 2.59 | 0.96 | 0 | 2.71 |
| | (0.9523) | (0.6646) | (0.6792) | (0.974) | | (3.31) | (3.31) | (4.77) | (9.2) |

Despite that IRM could be a good OOD algorithm, using IRM penalty as the metric to judge the OOD property of a learnt model still has much weakness, and some are severe. First, in different tasks, the value of $\lambda$ to obtain an OOD model may be different, so as other hyper-parameters like "anneal_steps" in IRM code. Without exhaustive search on the proper value of the hyper-parameters, it's easy that IRM overfits on the penalty term (which is the situation in VLCS). When IRM overfits, the IRM penalty will become quite small (higher $\lambda$ often leads to smaller penalty), but absolutely overfitting on penalty term will not result in good OOD accuracy. Therefore, the balance between loss and penalty is important. However, how to find a balanced point? This is a model selection problem, and Gulrajani & Lopez-Paz (2020) propose that an OOD algorithm without model selection is not complete. No matter what to be used as the metric, it cannot be IRM penalty since we cannot use what is included in the training process as the metric to select training hyper-parameters.

Second, IRM penalty shows a bias on different algorithms. In the Table 5, the IRM penalty of IRM is smaller than most algorithms. Besides, although the OOD accuracy of Mixup is similar to ERM, its IRM penalty is significantly higher. This is not strange but will limit the usage of IRM penalty. As for our metric, we mention that small $\mathcal{V}_{\gamma|\theta}$ is better. However, the understanding of "smallness" is based on the relative value of the shuffle version and standard version of $\mathcal{V}_{\gamma|\theta}$. As mentioned in section 5.2, when $\mathcal{E}_{all} \backslash \mathcal{E}_{tr} = \{S\}$, we can see that shuffle section 5.2 is obviously smaller than standard version in gDRO, but in ERM and Mixup, these value are relatively close or indistinguishable. In this case, we know that gDRO captures less invariant features and is not OOD

than the other two algorithms. During the whole process, we can circumvent the direct comparison of $\mathcal{V}_{\gamma|\boldsymbol{\theta}}$ in different algorithms, which is quite important. In summary, IRM penalty makes IRM a good algorithm, but using it as the general metric of OOD performance is completely another picture.

