# OpenReview forum: "Out-of-Distribution Generalization Analysis via Influence Function"
_ICLR.cc/2021/Conference — Reject_

### Official Review · AnonReviewer2 · 2020-10-15
**Interesting Idea - Experiments and Exposition Need Improvement**

**Rating:** 5
**Confidence:** 4

**Review:**

**OBJECTIVE**: Develop a method to reliably determine 1) if a model will generalize to out-of-distribution (OOD) samples 2) if it is advantageous to use algorithms that explicitly promote OOD generalization (in place of empirical risk minimization).

**CLAIMS**:
1)Test error can be a poor indication of OOD generalization.
2)Model stability across training environments (in the sense of “cross validation” between training environments) is a good indicator of OOD generalization.
3)The proposed index that uses influence functions to estimate how sensitive the model is to removing training environments does a good job in predicting how robust the model is against OOD samples.

**MAIN CONTRIBUTION**: The authors propose an influence function-based index to determine how well the model will generalize to OOD samples and whether OOD methods are needed in the first place.

(Note: PAIR stands for “please address in rebuttal”)

**STRONG POINTS**:
**Valuable line of inquiry**: The paper is on a very important topic that will be relevant to many in the ICLR community.
**Valuable connection**: The idea of utilizing influence functions to analyze OOD properties of machine learning models is, I believe, novel and interesting.
**More efficient than cross validation across training environments**: Assuming that influence functions are accurate enough to estimate the effect of removing training environments, the proposed method can significantly speed up the process of “running cross validation across different training environments”. Also, running the influence function computation on only the top model (i.e. the classifier) makes sense, and can significantly speed up computation – making the model feasible on even potentially large-scale tasks.


**WEAK POINTS**:
**Experiments**: Given that the main contribution is a heuristic measure of OOD generalization, I believe the experiments section is particularly important to prove that the proposed index is indeed useful for sufficiently complex, realistic problems (if not large scale). I believe the experiments are lacking in several ways:
1.	**Lack of different learning objectives**: There are many methods described in the related works section for learning models that generalize to OOD samples. However, only IRM and REX are used in the experiments.   I consider, in particular, the lack of “the robust learning objective” important. This seems particularly important to me, as I expect the proposed index to favor models trained with the robust learning objective that actually might not generalize to OOD samples. (PAIR)
2.	 **Lack of benchmark OOD generalization diagnostics**: Are there other papers out there that try to quantify how well a model will generalize to OOD samples? How does the model compare against those? For example, the paper “In Search of Lost Domain Generalization” by Gulrajani and Lopez-Paz (2020)[1] (which is not cited), and its related works section, seems quite relevant to this paper and probably should be considered by the authors – both in terms of which benchmark datasets are suitable, and which model selection criteria to use. (PAIR, if you have the time and resources) It would also improve the paper if the authors could try actually running some kind of “cross validation across training environments” (which the proposed approach is, as far as I understand, aiming to more or less capture more efficiently) to see how well it compares with the proposed approach? (PAIR). The authors also note that the proposed approach can be used to see if different training environments are actually different enough to require OOD generalization approaches for training. Couldn’t one simply check this by training a classifier to predict which training environment each example belongs to? If the classifier achieves random accuracy, one probably doesn’t need anything more than ERM. Am I right by saying this? (PAIR)
3.	**Lack of more realistic datasets**: All of the experiments in the paper are run on highly synthetic and unrealistic datasets. Although these experiments are still extremely helpful, it would improve the paper drastically if there were experiments run on more realistic datasets. Again, In Search of Lost Domain Generalization” by Gulrajani and Lopez-Paz (2020) seems like a valuable reference.  (PAIR if you have the time and resources)

**Faithfulness of influence functions**: Influence functions, in essence, try to estimate how much the weights would be perturbed if the training set didn’t have particular training examples (or environments in this case). The experiments reported in the paper are small-scale enough that one can likely afford to actually remove particular environments, retrain, then compare the weights with what’s predicted by the influence function. Did the authors run such experiments? (PAIR) I realize that the authors cite Koh (2019) that shows influence functions can still be useful to predict the effect of large number of training examples, but I believe the current setting is different enough that it would be worthwhile to repeat such experiments.

**Lack of justification for the particular form of the index**: It’s not that clear from the paper why the authors picked the particular form of the proposed index. Did the authors consider other forms? For example, I wonder if one could alternatively use the gradient of the excluded training environment with respect to the perturbation (which can be efficiently computed using influence functions) to derive an alternative index to compete with the current one.

**Is test accuracy really that problematic?**: I agree that without plenty of test environments, simply relying on test accuracy is problematic, as the authors point out with a concrete example. However, couldn't a sufficiently rich test set be seen as the gold standard to test OOD generalization, as also expressed in [1]?

**Missing related work**: I believe [1] is important to cite in this paper.

**DECISION**: Weak Reject.
Main reason: While I find the paper interesting and the line of inquiry promising, I believe it is not yet ready for publication, especially due to the weakness of the experiments section. With a better selection of baselines and benchmarks (both in terms of model selection criteria, more OOD generalization approaches (like robust learning objective) and more justification as to why the current form of the index is selected, the paper, in my opinion, can be ready for publication.

**QUESTIONS TO AUTHORS**: In addition to the sentences I’ve marked with the acronym “PAIR”, I have the additional questions:
1.	What do you mean by “same model change” in the last paragraph of page 4? Lets say I combine MNIST and CIFAR10 images to create a 20-way classification dataset, and I’m viewing MNIST and CIFAR10 as different training environments. Why should the classification model respond the same way to perturbations made to the different environments in this case?
2.	Just to clarify: does “2-norm for matrix” refer to the Frobenius norm, or actually the operator 2-norm?


**MINOR POINTS**: (these didn’t play much role in my decision)
Unpolished writing: I‘d recommend the authors to carefully go through the paper to remove as many grammar errors and typos as possible. That being said, the existence of these errors didn’t make it harder for me to read the paper, hence didn’t contribute to my final decision (at least consciously).
1. You might also consider restructuring the paper to make the introduction less meaty, and dedicate a section to discuss why test accuracy is limited.
2. You might want to describe what influence functions are verbally first, before giving a formula for it.
3. In section 3.2, you use both IF and I to mention influence functions, as far as I can see.
4. It would be nice if Table 1 refers to Section 5.1.

[1] In Search of Lost Domain Generalization, Ishaan Gulrajani and David Lopez-Paz∗  2020

---

> ### Author Response · Authors · 2020-11-17
> **Response**
>
> Thank you for your constructive comments. There are some important points which we hope to clarify and address here and in our revision.
>
> 1.  "Lack of different learning objectives: ......" and "Lack of benchmark OOD generalization diagnostics ...... to see how well it compares with the pro-posed approach?"
>
> **Response:** Thank you for the comments. I would like to answer these two questions together.   Gulrajani and Lopez-Paz (2020) implements 14 algorithms with consistent experimental conditions on 6 domain generalization datasets and systematically compares ERM with the domain generalization methods. This work is important and is related to this paper. We will cite this paper in the revised version. On the other hand, we should notice the difference between the different domain generalization tasks. Denote $\mathcal E_{tr}$ as the set of training domains and $\mathcal E_{all}$ as the set of all possible test domains. We assume $\mathcal E_{all}$ is unseen. Consider the following setups: (1) optimize the average performance over $\mathcal E_{all}$; (2) optimize the performance on the worst-domain in $\mathcal E_{all}$; (3) $\mathcal E_{all}$ is disjoint to $\mathcal E_{tr}$; (4) $\mathcal E_{tr}$ is a subset of $\mathcal E_{all}$. In this paper, we consider the problem (2) + (4), while Gulrajani and Lopez-Paz (2020) consider (1) + (3). Let us take PACS for an example to illustrate the difference. In Gulrajani and Lopez-Paz (2020), the experiments are training on three domains and testing on the fourth one. Then the transfer accuracy is used to represent the accuracy of the domain generalization problem (1) + (3). For the problem considered in our paper, this evaluation procedure is incorrect. We should train on three domains and test on all four domains. Then the worst accuracy on the four domains measures the OOD performance.
>
> We review the methods designed for the setup (2) + (4) and select IRM and REx to check our metric. Our method is derived from influence function, which is a tool from robust statistics. It checks the stability of the top model given the learnt representation $\Phi$. I agree that our metric may favor models trained by robust learning objective. According to our preliminary results on VLCS, our metric prefers ERM (metric mean: 1.82, var: 0.42), and then Mixup (metric mean:2.68, var:0.10) and groupDRO (metric mean: 3.33, var:0.10). Our metric on IRM is much larger than that on ERM, Mixup and groupDRO.
>
> As to "cross-validation across training environments", Figure 1 gives an example on Colored MNIST that the cross-validation over three domains cannot identify the OOD models.
>
> 2. "The authors also note that the proposed approach can be used to see if different training environ-ments are actually different enough to require OOD generalization approaches for training. ...... Am I right by saying this? "
>
> **Response:** Thank you for this comment. You are right. Your method checks whether a classifier can distinguish the training environments. Intuitively, we intend to check whether the learnt model treats all training domains equally. Thus we employ the proposed metric.
>
> 3.  "Lack of more realistic datasets: ......"
>
> **Response:** Thank you for this suggestion. Recall our answer to Question 1. The problem we considered in this paper is different from the problem in Gulrajani and Lopez-Paz (2020). It is not easy to check the OOD accuracy under the settings (1) + (3). For the common-used datasets, e.g. PACS, VLSC, the number of domains is limited which cannot provide strong experimental evidence to support that one method has better OOD accuracy. However, if the causal relationship is clear, e.g. Colored MNIST, the evaluation becomes easier. We can point out the worst domain according to the causal relationship. Thus we only consider the bayesian network and Colored MNIST in this paper. On the other hand, we have used our metric to measure the diversity of domains in VLCS. For ERM on VLCS, the mean and variance of our metric are 1.8228 and 0.4235 respectively. While for ERM on shuffled VLCS, the mean and variance of our metric are 2.3687 and 0.8836 respectively. Thus the VLCS dataset does not show enough diversity under our metric, which implies that an OOD method may not be needed.
>
>
> ------ Reference ------
>
> Gulrajani, I., & Lopez-Paz, D. (2020). In search of lost domain generalization. arXiv preprint arXiv:2007.01434.

---

> > ### Comment · AnonReviewer2 · 2020-11-18
> > **Thank you for your response**
> >
> > Thank you for your response.
> >     1. Discussion on Gulrajani, Lopez Paz (2020): I might have misunderstood something: did you mean to say, in (1) and (2), that the goal is to optimize performance on $\epsilon_{tr}$, instead of $\epsilon_{all}$? Regardless, I agree that the two papers compute the final test loss differently. The main reason I mentioned that paper is that the training setup in that paper is close-enough that you can still probably make use of both the algorithms and the datasets investigated there. For example, they also consider IRM and MixUP, and many more, including DANN which seems like a popular baseline this paper can consider.
> >     2. Having limited number of domains makes testing difficult: Unfortunately, in many academic and real world problems, we don’t have access to a large number of domains. Therefore, I believe the requirement of having many domains for the method to be practical is an important limitation. To make the paper stronger, I believe the best options are 1) to bite the bullet and still report results on datasets with a small number of domains (such as VLCS, PACS and others) 2)provide more theoretical justification for the proposed metric, or 3) explain in sufficient depth, in the main body of the paper, why your method is not applicable to problems where there are not that many domains. Perhaps your metric (or variations on it) can still be useful in such settings, if the domains are sufficiently distinct from each other? Maybe you can run some experiments on the synthetic datasets you consider on how much the number of domains affect the efficacy of your proposed method?
> >
> > I should note that I find the proposed method interesting and novel. My central critique is that since the main contribution is a heuristic, there should be more convincing evaluation to back up the claim that the proposed heuristic is useful and practically relevant. Results on ColoredMNIST and synthetically generated Bayesian Networks is valuable, but not sufficient in my opinion. This is why I currently choose to maintain my score.

---

> > > ### Author Response · Authors · 2020-11-24
> > > **Reply**
> > >
> > > We genuinely thank you for your interest in our work and your detailed review. We notice that you mention many comparisons that should be done. To see how our metric performs in real datasets, we conduct lots of experiments in the last two weeks, and we have added our result to the main text of our paper. While we cannot run too much more experiments due to the limitation of GPU resources, we believe our supplemental experiments in VLCS is enough to show the performance of our metric and to reply most of your question. Specifically, we consider a real scenario and show step by step how we use the metric. Please refer to the third part of the experiment section.
> > >
> > > Besides, for the question not mentioned in the section, we would give our reply here.
> > >
> > > 1. **The goal**
> > >
> > > The goal is a model that achieves high OOD accuracy, i.e. $\min Acc^S, S\in\mathcal E_{all}$. However, due to the limitations, we can only get $\mathcal E_{tr}$. So the goal is to use the information provided only in $\mathcal E_{tr}$ to achieve high OOD accuracy.
> > >
> > >
> > > 2. **Number of domains**
> > >
> > >
> > > We notice that you mention the limitations of few available domains, i.e. #$\mathcal E_{tr}$ is not large. Indeed, this is exactly where we are, and exactly the situation where our metric should be helpful. After all,  if $\mathcal E_{tr}$ is large and diverse enough, for any algorithm $\mathcal A$ like ERM, the goal of $\mathcal A$ may be similar to the goal of OOD accuracy. Hence $\mathcal A$ performs well. In those cases, if we compare the shuffle version (Section 4.2) and none-shuffle version of our metric, we will find them similar.
> > >
> > > However, when $\mathcal E_{tr}$ is not large, we argue that our metrics can still be helpful. We have reported the results in VLCS, but we would also like to give some insights. We suggest that the OOD property of domains is not a model-irrelevant issue. Instead, the model should be taken into account. For example, for a model which consistently outputs $0$, all the domains are the same, since no feature is used. Under this model, our metric is very small. But it doesn't harm our argument. As we clarified in the paper, our metric is a supplement of test accuracy. In the above model, its test accuracy is low, and there's no need to consider out metric. This example tells us the usage of the metric from the opposite side: when the model does achieve no bad test accuracy in $\mathcal E_{tr}$, can this excellence be seen in $\mathcal E_{all}$ as well? To answer this question, we assume that, with similar test accuracy, the model which use less varying features in $\mathcal E_{tr}$ and more invariant features should be more OOD to $\mathcal E_{all}$. Of course, according to the "No Free Lunch Theorem", this may not be true. However, in section 4.2(OOD Model), we discuss this problem and summarize that demanding a model to avoid features that are invariant in $\mathcal E_{tr}$ but varying in $\mathcal E_{all}$ is unrealistic.
> > >
> > > If we admit that with high test accuracy, using more invariant features and less varying features will make a model more OOD in unseen domain $\mathcal E_{all}\backslash \mathcal E_{tr}$, then the problem becomes: whether our metric can capture this property of the model. Here we introduce the influence function, which is a well-established tool in Statistics, to play the key role in our metric. Using the influence function, we can capture the change of $\gamma$ when we enlarge different domains, and this change can be rigorously proved (in Appendix A.2), so long as the learnt model can arrive a point $(\beta,\gamma)$ with $\nabla_{(\beta,\gamma)}\mathcal L(f,\mathbb S)$ is closed to $0$. Based on the influence function, we consider the difference of $\mathcal {IF}$ using the 2-norm of its covariance matrix and apply a monotonic transformation $\ln(\cdot)$ to make it more distinguishable. So long as the influence function can capture the change of $\gamma$, our metric should capture whether the learnt model $\hat {\mathbf \theta}$ learnt from algorithm $\mathcal A$ with high test accuracy regards different domains as the same (learning invariant features) or not (using varying features). The practical use of our metric in VLCS shall help demonstrate our point.
> > >
> > > Above all, we sincerely appreciate your devotion to our work. We hope the supplemental experiments can answer your question and dispel your doubts!

---

### Official Review · AnonReviewer3 · 2020-10-27
**Paper proposes to use influence functions to build a new metric to understand OoD performance from training data since accuracy is not a good metric. This issue was already resolved in the existing works. Many things incorrect with the paper including a major mistake in using the regularizer from the work of Arjovsky et al.**

**Rating:** 4
**Confidence:** 5

**Review:**

Summary

The authors study the problem of out-of-distribution (OoD) generalization. The key question authors seek to answer is when given access to data from multiple training environments, can one only rely on test accuracy? or does one have to rely on some new measures to estimate the out-of-distribution performance of the model. The authors develop a metric based on influence functions, which authors claim is a better reflection of OoD accuracy than test accuracy. The metric proposed by the authors measures the variance in the model when the data from each environment is upweighted. The authors show that the proposed metric empirically correlates to the OoD performance of the models.

Pros:
I appreciate the problem that the authors consider. It is important to develop metrics that are good indicators of OoD performance since accuracy on train environments is not a good indicator.

Cons:
I divide my concerns into different subsections below:

a)  IRM Arjovsky et al. (2019) already solve the problem that authors want to solve in this work:
The authors claim that they show that how test accuracy is not a good metric and highlight the need for a new metric to measure OoD performance. IRM (Arjovsky et al.) have already established this point. Recall the colored MNIST experiment in Arjovsky et al., which has precisely already shown accuracy over any training environment does not suffice in selecting a predictor with good OoD behavior.

If not accuracy, then what is the right metric? The authors in the current paper propose an influence function-based metric but completely ignore the metric that is already used in IRM to inform the learning of the OoD model. After all IRM is able to learn a model based on training environments itself, which means the IRM loss itself can be thought of as a proxy for the OoD performance. Recall that the loss used in IRMv1 in Arjovsky et al. balances the accuracy over the training environments and invariance achieved across all of them.

Let us now precisely state a metric based on the work of Arjovsky et al.  We perfom the following steps:

i)      Split the data in each environment into a train split and a validation split. Learn the model using the train split from each
environment.

ii)     Use the trained model and compute the IRMv1’s penalty value on the validation split for each environment

iii)	Compute the mean and the variance of the penalty value across environments.

We now discuss why the above metric can capture the OoD performance better than just training accuracy for instance.
The penalty in IRMv1 checks if the predictor learned satisfies invariance conditions across all the environments. The penalty by definition is greater than zero. If the penalty is close to zero for all the environments, then the model satisfies invariance across environments.

Consider two models: one trained from ERM and other trained from IRM. Assume that they have a small average validation error across all the environments. Let us analyze the following scenarios:

i)	Suppose the mean of the penalty value across the environments is close to zero for the IRM model. Suppose the mean of the penalty value across the environments is much larger than zero for the ERM model. In such a case, the IRM model satisfies invariance and will perform better than ERM model on OoD data.

ii)	Suppose the mean of the penalty value across the environments is close to zero for the IRM and the ERM model. In such a case, it is likely that both ERM and IRM models have a good performance on OoD data.

iii)	Suppose the mean of the penalty value across the environments is similar for both the IRM and the ERM model but is greater than zero. In such a case we cannot rely on the mean, we should use the variance in the penalty value across environments. The model with a lower variance is likely to have a better OoD performance.


b)	Incorrect use of IRM loss to compute influence

The authors define in equation (3) a regularized loss function. Following equation (3) the authors define two types of regularizers from IRM and REx. The regularizer for IRM is incorrectly defined as \sum_{e}(\|\nabla_{w}\ell(g(w \Phi), S_e)\|)^2. The correct regularizer is
\sum_{e}(\|\nabla_{w}\ell(w \Phi, S_e)\|)^2. There is no function g needed as the scalar w serves the purpose for the classifier that operates on top of the representation. Subsequently everything authors do with computing changes on g is incorrect as g itself is not needed.


c)	Concern regarding influence computation in REx model

The authors divide the predictor into a representation and a classifier on top. However, it is not clear why would one do that for REx. REx uses the entire predictor as one. Hence, breaking down the predictor in the context of REx and analyzing the influence on the classifier does not make sense. There needs to be a better justification for doing what authors seem to be doing.



d)   Influence metric should be defined properly

The authors want to analyze influence of all the points in a given environment. The authors suggest upweighting the points but for reasons not very well explained the authors decide against upweighting the points inside the penalty. The whole principle of computing influence of a point is to upweight that  point in the entire loss and then determine how does the trained model change. The current approach decides only to upweight the part concerned with ERM loss and not the regularization.
The penalty is a crucial part of IRM and not upweighting it seems to against the spirit of understanding the impact of each environment on the model.

e) What is the correct way of computing influence of a each environment?


I would suggest the authors to follow the standard steps used in Koh et al. and upweight the points in an environment in both the ERM part of the loss and the regularizer. Instead of analyzing the impact on some model g, the authors should see the impact on the model phi itself. Analyzing the variance of change in the model phi may be correlated with OoD. However, this remains to be seen.

f) Concern with using influence at all

The main reason to use influence functions is the computational advantage offered. When we do not want to retrain the model for every point we want to analyze, it makes sense to compute the influence. However, the authors are computing the influence per environment. In general, we do not expect too many environments, thus why not just retrain the model after upweighting the points in an environment. The only argument that can be made in favor of deriving influence in this case is if the authors subsample from the environments and determine the distribution of influence values. However, authors have not discussed any such justifications.

Recall from Koh et al., the analysis of influence requires that the number of points being upweighted be much smaller than the total number of points.  An entire environment can constitute a significant proportion of the total data and changing all the points in an environment can lead to severe miscalculation of influence. The authors should have shown through a plot similar to Figure 2 in Koh et al. that computing influence of an environment vs. actually retraining the model lead to similar values.

g) Concerns regarding experiments

If the authors believe that influence based metrics do lead to a better estimate of OoD performance, then they should have at least compared with metrics based on IRM penalty that I describe above and also REx penalty. All the IRM based methods rely on some type of signal from the training data to train a good OoD model.

Quality:
Unfortunately, the paper is of poor quality. The authors have not carefully analyzed how existing works already address this problem.

Clarity:
The writing of the paper is average. An explanation of how influence is computed for REx vs IRM vs ERM should have been clearly stated and if they used the same technique, then an explanation of why should have been there too.

Significance:
The problem authors are trying, i.e., building metrics that are better indicators of OoD performance  than training accuracy, is very important. However, the approach taken by the authors is incorrect and ignores many important comparisons.

---

> ### Author Response · Authors · 2020-11-17
> **Response**
>
> Thank you for your careful review and comments. To our best understanding, a few concerns seem to arise from misinterpreting our paper’s content: we apologize if our manuscript has not been more clear and might have caused those confusions. We hope our point-by-point response below can address your concerns.
>
> a1) IRM Arjovsky et al. (2019) already solve the problem that authors want to solve in this work ...... does not suffice in selecting a predictor with good OoD behavior.
>
> **Response:** The problem considered in the present paper is that the test accuracy on a test domain is not enough to measure OOD performance. Let us take PACS for an example to illustrate this problem. The common-used experiments are training on three domains and testing on the fourth one. Then the transfer accuracy is used to represent the OOD accuracy. This evaluation procedure is incorrect and may fail to measure the OOD performance. This is not the problem specified by the Colored MNIST experiment in Arjovsky et al. (2019). In the introduction, we have presented a cross-validation example in which three domains of the Colored MNIST (color flipping rate: 0.0, 0.1, 0.2) are available. Figure 1 shows that the cross-validation fails to identify an OOD model. Back to the IRM paper, the results of the colored MNIST experiment is convincing because the spurious feature (color) is known such that one can select the domain with color flipping rate 0.9 for testing. If the causal relationship is unclear and the access to domains is limited, e.g. the cross-validation example, it is impossible to implement the test on the worst domain. Hence the test accuracy is a biased estimate of the OOD accuracy.
>
> a2) "If not accuracy, then what is the right metric ...... We perform the following steps ......"
>
> **Response:** It is a good question. First, the influence function-based metric is supplementary to the accuracy metric and identify whether the learnt model treat the training domains equally. Intuitively, the high test accuracy means the model learns some features related to the target label. In addition, the influence function-based metric checks whether the learnt features are stable or invariant over the training domain. Second, we want a fair metric. We believe that the IRM penalty can derive a metric that guides the tuning of IRM. Our preliminary experiments show that the IRM penalty is more friendly to the models learnt by IRM while the REx penalty prefer the models learnt by REx. Thus we turn to the influence function, which is a well-established tool in robust statistics. Third, it is not easy to explain the IRM penalty metric. According to the IRM paper, the IRM penalty should be able to measure some types of causality. However, it lacks a precise explanation. If simplify this metric as a robustness metric, we cannot explain how IRMv1 discovers causal features by penalizing a robust quantity.
>
> b) Incorrect use of IRM loss to compute ......
>
> **Response:** Sorry for the typo. The regularizer for IRM should be $\sum_{e}(\big|\nabla_{w}\ell( w g(\Phi), S_e)|_{w=1.0}\big|)^2$. The IRMv1 uses $\Phi$ to represent the entire predictor which is $g(\Phi)$ in this paper. Throughout this paper, the notations are consistent. Notice that w is a scalar and $\Phi$ stands for the features. For binary classification, a scalar $w$ can serve the purpose for the classifier that operates on top of the representation. For more general cases, the function $g$ represents the top model and is needed.
>
> c) Concern regarding influence computation in REx model:......
>
> **Response:** We cannot agree with this comment. IRMv1 also use "$\Phi$" to represent the entire predictor ("$g(\Phi)$" in our paper) and does not divide the entire predictor into a representation and a top classifier. According to your comment, breaking down the predictor in the context of IRMv1 does not make sense.
>
> d) Influence metric should be defined properly....
>
> **Response:** Thank you for this comment. The total loss includes the ERM loss and a regularization term. The ERM loss depends on the sample size of each domain.  The regularization term encourages some type of equality among domains. So the regularization is a domain-level penalty and should not depend on sample size. Hence we only upweight the ERM loss.
>
> e) What is the correct way of computing influence of each environment?
>
> **Response:** Thank you for the suggestion. This paper considers the robustness of the top model $g$ given the features $\Phi.$ I think we have given a correct way to compute influence under this setup.
>
> f) Concern with using influence at all
>
> **Response:** Please refer to Koh et al. 2019 which shows that influence function can be used to approximate the group effect.
>
> g) Concerns regarding experiments
>
> **Response:** Please see my answer to a1) and a2).
>
>
> ---Reference---
>
> Koh, P. W. W., Ang, K. S., Teo, H., & Liang, P. S. (2019). On the accuracy of influence functions for measuring group effects. NeurlPS

---

> > ### Comment · AnonReviewer3 · 2020-11-19
> > **Response**
> >
> > Hi authors
> >
> > Thanks for your response.
> >
> > 1.  It is not clear based on your response why IRM penalty cannot be used as a proxy to measure OOD performance in the manner I described above. It should be a very good proxy and comparison benchmark. If you would do extensive experiments and show that IRM penalty does not capture OOD performance as well as your metric, then it would be interesting. I would like you to take this as a constructive suggestion to carry out such comparisons with both IRM penalty and REx penalty.
> >
> > 2. I do not still get how the typo (which by the way is a very big typo) addresses the whole problem.
> >
> >     i) Say w(g(\Phi)) is the new predictor, w acts as classifier on top, Phi is representation, what is g doing in the middle as it just seems a redundant part of the representation. If you say g is a classifier, then the gradient should also be w.r.t the parameters of g, which is not what IRMv1 does.
> >
> >     ii) When you compute influence you seem to do it on parameters of g which according to me as a part of the representation.  To put it differently. I would consider g(\Phi) as just another function Phi' parameterized by gamma and beta. When you compute influence you are actually computing influence on parameters of representation and not the classifier. Do you agree?
> >
> >     iii) If you think g is a separate classifier and w is another classifier that acts on top, then that is simply not what IRMv1 does. I fail to understand why would you not follow the exact IRMv1 setup and instead introduce a confusing middle layer with g.
> >
> >
> > 3. I am saying you are breaking down a predictor into a top model and a representation for REx. REx does not break down the model like this. Please note that my comment is not about IRMv1.
> >
> > 4. I have seen the work of Koh et al. What I am trying to say is observe that the group or environment in your case is so big and as a result the Taylor approximation at the level of groups would not hold. Would you not agree that it would make sense to just upweight the data and rerun the experiment and run a comparison with influence computation. Also, since there are not so many environments after all why do any influence computation as you could just upweight and rerun.

---

> > > ### Author Response · Authors · 2020-11-24
> > > **Reply**
> > >
> > > Thanks for the detailed comments.
> > >
> > > **Q1.** “It is not clear based on your response why IRM penalty cannot be used as a proxy to measure OOD performance in the manner I described above. ...”
> > >
> > > **Ans:** We think that "`IRM could be a good OOD algorithm" is quite different from that "``IRM penalty is a good metric''. To see this, we analysis the performance of IRM penalty and our metric in VLCS. Please refer to experiment 5.2 and appendix 6.2 to see why this will happen. We also show in this two part how our metric works. We hope this may help us clarify why we prefer a new metric instead of using the peanlty of IRM or REx.
> > >
> > > **Q4.** "I have seen the work of Koh et al. ... ... Also, since there are not so many environments after all why do any influence computation as you could just upweight and rerun."
> > >
> > > **Ans:** We would like to answer this question first. We again emphasize that, we are not using influence function to emulate something. Instead, the calculation results is exactly what we need, i.e. when upweight a loss from $\mathcal L$ to $\mathcal L + \delta l(\mathbb S^e)$, what is the change of $\gamma$. That is to say, with function $g(\delta) = \arg\min_{\gamma} \mathcal L+ \delta l(\mathbb S^e)$, we are calculating $\nabla_{\gamma} g(0^+)$. Therefore, we are not estimating anything, e.g. estimating the change of $\gamma$ when we removed the environment. Instead, the derevative is what we want, and we prove the correctness of the calculation in the appendix.
> > >
> > >
> > > **Q2+Q3.**  "I do not still get how the typo (which by the way is a very big typo) addresses the whole problem..." and "I am saying you are breaking down a predictor into a top model and a representation for REx. REx does not break down the model like this. Please note that my comment is not about IRMv1."
> > >
> > > **Ans:**  Actually, we are not breaking down a predictor into a top model and a representation merely for REx. Instead, for most DNN, their will be a fully connected network (FNN) at the end, and this is the classifier. For example, on computer vision field, a typical CNN consists of many convolution layer (this is extractor $\Phi$) and a FNN (this is classifier $g(\cdot)$. The breaking down of $g$ and $\Phi$ is not only used in IRM. In reality, most DNN (at least in computer vision) has this structure, i.e. $f(\cdot) = g(\Phi(\cdot))$. During the whole process, there is nothing about OOD algorithm.
> > >
> > > Now lets come back to specific algorithm. In IRM, the penalty term in practice is defined as IRMv1. Here, IRMv1 regards the model as a whole, i.e. as $f(\cdot)$ and directly calculate $\sum_{e \in \mathcal E_{tr}} \| \nabla_w \ell\big(w f), \mathbb S^e\big)\big|_{w = 1.0}\|^2$. In Rex, the penalty consider nothing about the structure of $f(\cdot)$ as well. Therefore, out breaking down of the model is not from IRM, but instead it's just a popular structure.
> > >
> > >
> > > Now, whatever algorithm, given a loss function $\mathcal L$, we calculate the influence function value metioned above. You are asking that this is not the thing done in Koh et al., since when upweight an environment, we do not upweight the penalty term. We argue that, we are not emulating the effect of remove a group of point just as Koh et al. has done. Instead, we use the techique of influence function, can ``define'' what should be calculated ourself. As for why we do not upweight the penalty, we should say that what we care is the usage of features, and this is irrelevant to the penalty term. That is to say, we consider to directly check the invariance of the top model.
> > > In Section~3.1 of the revised version, we have added more explanation about these two methods. We hope our answer and the revision can address your concerns.
> > >
> > > Above all, from the bottom of our heart we appreciate your helpful comments. We hope that our response will clear up your doubts.

---

### Official Review · AnonReviewer4 · 2020-10-28
**Using influence functions to understand out-of-distribution generalization**

**Rating:** 4
**Confidence:** 4

**Review:**



=================================

**Update after discussion period**

My feeling is still that the proposed method has a lot of interesting potential, but that the paper still needs some improvement. I've bulleted my main remaining concerns below:

1. The clarity of the writing still needs to be improved in many places. Notably, the authors' discussion of their metric is mostly discussion-based (rather than providing concrete theoretical claims about their metric and its strengths). While this isn't necessarily a problem, such a presentation really needs precise language and phrasing so that the details of the claims and their supporting arguments can be completely understood. From what I can follow, I think many of the authors' arguments are going in good directions, but the writing should be improved to be sure.
2. I still don't completely understand why $\beta$ is being fixed here. The argument in Appendix A.3 seems to say that if $\hat\beta$ varies, then the meaning of $\hat\gamma$ changes, and so looking at the change in $\gamma$ would not be meaningful. I agree, but that does not mean we cannot look at the combined change of $\beta$ and $\gamma$. It seems like this would be straightforward to experiment with on some smaller problems that only have parameters in the tens of thousands; I think it would be better to include such experiments rather than trying to argue verbally that such an approach will not work well.
3. I am also not sure I see why we should not be interested in leave-one-domain-out CV. I appreciate the clarification that Figure 1 is actually showing CV's estimate of error, and definitely find this to be a compelling experiment. It is pretty surprising to me that the authors' metric could succeed in this case, given that the influence functions are an approximation to the parameter change as each domain is left out (this is true, whether or not this is the intent of the metric). I think further investigation of this point is needed. Would computing the proposed metric with the parameter changes under leave-one-domain-out CV not detect the OOD issues in Figure 1? Or is it just that feeding these parameter changes into the function measuring test error that is providing a poor assessment of OOD generalization in Figure 1?

On a different note, I appreciated the increased discussion of "the shuffle;" I think this is an important part of the paper that didn't come up much during the discussion period. As a side note, I think it is more common to just call this sort of thing a permutation test. And, along those lines, it would be good to actually perform a permutation test (i.e. run over many shuffles and examine the distribution) so that we know for sure the reported shuffles aren't just unluckily high/low.

===================================

**Original Review**

This paper is focused on understanding whether a model fitted model is going to have out-of-distribution issues. In particular, the authors consider models trained on multiple "environments," which may fail to generalize to future environments. To detect whether a model will fail in this way, the authors propose to look at the change in model parameters as each environment in the training set is infinitesimally upweighted (i.e., they compute influence functions). If upweighting each environment the same amount changes the model by a similar amount, the authors reason that the model must not be overly fit to any one environment, and so will generalize to future environments well.


I think a major issue is the derivation of Eq. (4), which claims to be the "change of top model g caused by upweighting the environment." I'm interpreting this as the change in the parameters $\gamma$, which are part of the overall parameter vector, $\theta = (\gamma, \beta)$. My understanding of how to calculate influence functions is that the influence function for the overall parameter $\theta$ would be:
$$
    -
   \begin{pmatrix}
            H_{\gamma\gamma} & H_{\beta\gamma} \\\\
            h_{\gamma\beta} & H_{\beta\beta}
    \end{pmatrix}^{-1}
     \begin{pmatrix}
          \nabla_\gamma f \\\\
          \nabla_\beta f
     \end{pmatrix}.
$$
If $\gamma \in \mathbb{R}^D$, I would say that the (infinitesimal) change in $\gamma$ is really the first $D$ entries of this vector. But this is not what is written in Eq. (4), and the derivation of Eq. (4) in Appendix A.2 does not actually derive Eq. (4) (the derivation in the Appendix derives the equation written above and does not refer to $\gamma$). I think this is especially important, as the reason that Eq. (4) is computable is that the matrix to be inverted is $D \times D$, rather than the much larger $dim(\theta) \times dim(\theta)$ in the equation I've written above. Can the authors clarify what's going on here?

The clarity of the writing is also an issue. Many sentences have grammatical issues (e.g. "...according to detailed derivation in Appendix A.2, the change of top model g..." is missing the word "the" twice). Most of these don't make the paper any harder to understand, but should still be fixed. I did have trouble following some of the more expositional parts of the paper. For example, the following sentences weren't clear to me:

>First, not all OOD problems demand models to learn invariant
features, e.g. when varying features are always more strongly correlated to the prediction. But to
our concern, we regard the OOD problem as a bridge to a truly causal model, so these minor ”OOD”
problems are out of our consideration. To a large extent, invariant features are still the major target
and our proposal is still a good criterion to model’s OOD property.

Could the authors provide more details about what their example in the first sentence refers to (is it a counterexample to a specific definition of an "OOD problem")? What is a "truly causal model" (one that only uses features that have a causal effect on the outcome)? Is this a different goal than building a model that uses the "invariant features" discussed in the third sentence?

A final issue is that the authors' method seems close to cross-validation (CV), where each fold of CV leaves out an environment, but this connection isn't discussed. On the surface, it seems like an even better way to understand how much a model is depending on an environment is to actually remove the environment from the training data, rather than just changing the weight of the environment by an infinitesimal amount. Could the authors describe why their method is more desirable than CV in this context? If the issue is computation time, note that influence functions have been used to approximate CV [Giordano et. al 2019].

----------------References---------------------
R. Giordano, W. T. Stephenson, R. Liu, M. I. Jordan and T. Broderick. A Swiss army infinitesimal jackknife. AISTATS. 2019.

---

> ### Author Response · Authors · 2020-11-17
> **Response**
>
> Thank you for your insightful comments. We hope our point-by-point responses below address your concerns.
>
> 1. I think a major issue is the derivation of Eq. (4), which claims to be the "change of top model $g$ caused by upweighting the environment......."
>
> **Response:** Thank you for this comment. Sorry for the confusion. Hope the following specification can make up for our carelessness. Our value of interest is the change of $\gamma$, the parameters of the top model (classifier based on extracted features), with respect to upweighting a certain environment by a small $\delta$. Note that in our specification we assume fixed $\beta$, which implies the feature-extraction process is left unchanged. We elaborate this by clarifying the following points.
>
> Firstly, we are not emulating the change of $\gamma$ when the dataset is totally removed. Instead, we care about the change of $\gamma$ when we have a tendency to enlarge a dataset. Our key in-sight is that, if the changes of $\gamma$ when enlarging different environments are different, we know that under such loss function, the learnt model are constructed on varying features (which en-tails less OOD). Second, when calculating the change of $\gamma$, we assume fixed $\beta$, i.e. consistent feature-extraction process. This is not an ad hoc assumption, since if we allow for the change of $\beta$, the change of the use of these features are less informative. Suppose, for example, when enlarging an environment, the intensity of a feature decrease significantly, then the $\gamma$ corresponding to this feature may increase, while in fact the use of this feature decrease.  It would be erroneous if we allow for changes in $\beta$. Conditioning on constant $\beta$, it directly follows that the influence function of $\gamma$ is reduced to Eq. (4). In Appendix A.2, due to the simplicity of the model, $\gamma$ is identical to $\theta$ (i.e. the whole model are a classifier, and data itself is feature). So Appendix A.2 gives the derivation with respect to $\theta.$ In the revision, we will correct the notations in Appendix A.2.
>
> 2. The clarity of the writing is also an issue...... Could the authors provide more details about what their example in the first sentence refers to (is it a counterexample to a specific definition of an "OOD problem")?......
>
> **Response:** To our best knowledge, "OOD" (Out-of-Distribution) is a concept that includes a large number of problems which are caused by the mismatch between the training distribution and the test distribution. The problem we considered is given in Eq (2), Page 1. The notation $\mathcal E_{all}$  is the set of all target domains which include the training domains. If $\mathcal E_{all}$ is unseen and is a small set, the invariant features or the causal features are not needed. The "truly causal model" stands for the model that only uses causal features. Strictly speaking, an invariant feature over $\mathcal E_{all}$ is not a causal feature unless $\mathcal E_{all}$ is sufficiently large.
>
> Let’s further consider how our proposed metric help us. When the value in ERM is small, we can conclude that the variance between training environments are small. In this case, if $\mathcal E_{all}$ is sufficiently large, we think the only way to improve the performance is to collect more environments, until training environments set $\mathcal E_{tr}$ can somehow reflect some diversity of $\mathcal E_{all}$. However, when $\mathcal E_{all}$ are not so large, guided by our metric, we think a OOD algorithm is not needed.
>
> On the other hand, if the value of our metric is large, we know that the model are using some varying features across $\mathcal E_{tr}$. Such a model can never be a truly causal model. In this cases, there are reasons to believe that the model will not perform well in $\mathcal E_{all}$, since it already uses varying features and such usage is large (be-cause the metric is large). To obtain better OOD accuracy, varying features at least should be more excluded, and our metric serves as an indicator. If a model eventually obtain small $\mathcal {V}$, is it necessary a causal model? No. However, at least this is already a step forward.
>
> 3. A final issue is that the authors' method seems close to cross-validation (CV), where each fold of CV leaves out an environment, but this connection isn't discussed....
>
> **Response:** The proposed metric measures the invariance of the top model and is supplementary to the test ac-curacy on test domains. Figure 1 shows that the cross-validation over three domains cannot identify the OOD models.

---

> > ### Comment · AnonReviewer4 · 2020-11-19
> > **Thanks for the reply**
> >
> > Thanks for the clarification on keeping $\beta$ fixed; that definitely makes sense with how Eq. (4) is derived in this case. I think this does ask the question, though, of what happens when both $\beta$ and $\gamma$ are allowed to vary. It seems like the input data $x$ can be considered “features” that the model is learning to use / over use just as well as $\Phi(x)$ can; that is, we might consider taking the covariance of the change of all of $\theta$ as our measure of OOD generalization. Is there some reason that we know fix-beta, vary-gamma (the proposal in the paper) will perform better than vary-beta, vary-gamma? As the authors point out, one is much more computationally expensive, but I think it would help to understand what tradeoff we’re making by making the (computationally necessary) decision to fix beta.
> >
> > I'm also not totally settled on the "leave-environment-out" cross-validation issue. I think I might be confused as to what exactly is being computed as “test accuracy” in the paper. It seems like the procedure being performed is: 1) train on all but one environment 2) report test accuracy as the average accuracy on all datapoints in the remaining environment. This is different from what I would think of as “test error,” which is to draw a number of new environments from $\mathcal{E}_{all}$ and average the error on these sets. This later idea of “test error” (which I suspect would more accurately reflect OOD issues) is what I would expect leave-one-environment-out CV to estimate. It seems that what is being reported as “test error” in the paper is the error estimated by one fold of leave-one-environment-out CV, whereas leave-one-environment-out CV would average across all of these folds.
> >
> > In any case, it would be very helpful if the authors could upload a revision addressing the clarity/grammar issues that are present in the submitted version.

---

> > > ### Author Response · Authors · 2020-11-24
> > > **Reply**
> > >
> > > Thanks for your response. We notice that there are two questions, about the choice of fixing or varying $\beta$, and about the cross-validation. We hope the following explanation could answer your questions.
> > >
> > > 1. **About fixing $\beta$ or not**
> > >
> > > Under our settings, input $x$ may come from different datasets, i.e. $x^e \sim P^e$ and $P^e$ are different. Therefore, if $x$ is a vector of features and $g(\Phi(\cdot))$ is the model built on these features, it's impractical to demand that the use of feature $x$ are invariant across domains. An example may help us clarify the problem. Say, our task is to predict the age of people from $S^1$ (Man) and $S^2$ (Female) and $x$ is the digit tensor of the photo. In order to capture the invariant feature to accurately project the age, a wise model should recognize the gender of the people and use the gender to eliminate the "systematic bias", e.g. to rectify some gender-based-feature to invariant ones. To do this, our feature extractor, a "CNN" model, for instance, will definitely utilize the varying features in $x$, if the whole model does want to eventually capture the invariant features.
> > >
> > > If we take $\beta$ into the calculation of influence function as well, the above process will result in a huge difference between the influence function of different domains, although the model is OOD as last. That is to say, requiring the model only to use the invariant part in $x$ is not that reasonable,  since $x$ is still to fine-grained and not abstract enough to be regarded as "features". Constraining ourselves only to use the invariant part in $\Phi(x)$, however, is more reasonable, since in classifier $g$ (with parameters $\beta$) it's time to make final predictions.
> > >
> > > We summarize that the fixing of $\beta$ is not a "trade-off". Instead, it's a "double-win" situation since it takes the necessity of being "varying" to different domain into consideration, in order to become invariant in the aspect of final features, i.e. $\Phi(x)$.
> > >
> > >
> > > 2. **About Cross Validation**
> > >
> > > Let's clarify the definition first: $\mathcal E_{all}$ is the set of all domains we care about, and OOD accuracy is the min accuracy in $S \in \mathcal E_{all}$. Suppose the subset of domains we can get is $\mathcal E_{tr}$. Whatever we use comes from $\mathcal E_{tr}$, including training, validation and test domains. Any attempt to get new $S \in \mathcal E_{all} \backslash \mathcal E_{tr}$ has been done early and is out of our consideration. In this case, suppose we choose a domain $S^{test} \in \mathcal E_{tr}$ as the test domain, then the test accuracy is the accuracy in $S^{test}$. According to Gulrajani and Lopez-Paz (2020), the leave-one method and the cross-validation method can be combined. For example, if we consider the cross-validation, then we should take turns to choose $S \in \mathcal E_{tr}$ as the test domain (which is called validation domain in that paper), and choose the average or min accuracy of $S\in\mathcal E_{tr}$ as the final metrics. In this case, test accuracy is defined identically as the average or min accuracy in $S \in \mathcal E_{tr}$.
> > >
> > > However, as we mentioned in the introduction, no matter which methods are used, it is still possible that the test accuracy cannot reflect the OOD accuracy, or can be inversely correlated to OOD accuracy. In the experiment in our introduction, no matter which domain is used as the test domain, the test accuracy is consistently high, even the minimum of cross-validation accuracy is high as well. However, as we point out, the OOD accuracy of the model is poorly low. The insight is that, if $\mathcal E_{tr}$ are only slightly different, the cross-validation may fail to help us judge the OOD property of the model.
> > >
> > >
> > > Thanks again for your kind comments. We will upload a revision that addresses the clarity and grammar issues. Besides, we do appreciate your heartfelt reviews!
> > >
> > >
> > > ------Reference------
> > >
> > > Gulrajani, I., & Lopez-Paz, D. (2020). In search of lost domain generalization. arXiv preprint arXiv:2007.01434.

---

### Official Review · AnonReviewer1 · 2020-11-10
**This work proposed a useful metric to decide when to apply IRM/tune IRM models - Why cant conditional mutual information based metrics be used ?**

**Rating:** 7
**Confidence:** 5

**Review:**

Summary:
     This paper tries to understand the following question - a) Given a set of environments - when would we expect an ERM solution to be completely bad in the out of distribution (OOD) generalization sense. b) Given a trained model, how can we certify that the model is good in an OOD sense when we dont have access to a new environment.

This metric might be useful to tune IRM procedures which has been identified as a real issue in some recent work.

IRM optimizes over representation $\phi$ and a classifier on top of it $\gamma \approx E[Y| \phi(X)]$ is remains the same across all environments.

 The metric proposed: Influence function which measures the change in $\gamma, \Delta \gamma $ caused by upweighting an environment $e$'s loss by $\delta$. The metric assumes that the IRM optimization is done using $\sum_e L(\gamma,\phi, S^{e}) + \lambda R (f, S^{e})$ where $S^{e}$ represents the dataset for environment $e$.

The limiting influence function when the weighing factor $\delta$ tends to $0$ is given by a inverse of a Hessian matrix of the regularized loss over all data  (whose changes are measured with respect to the classifier $\gamma$) and a gradient vector of the loss (without the regularizer) with respect to the $\gamma$ on the data of the environment which is up weighted.

Now, log of the spectral norm of the cross covariance matrix when changes are made to the different environments involving these influence measures is used as a metric.

To test a model: One has to check if this derived measure is low and test accuracy is high (test set made from training envrionments). If so then, that model can be expected to have a high OOD generalization to unseen environments.

To test environments if they need special OOD optimization frameworks: The authors propose to a) randomly create m environments, and b) use the pooled data and train an ERM model for both. If the metric proposed in b) is larger than the metric in a) across computed using the actual environments, the authors propose that one should check for an algorithm (like IRM) that is specifically tailored towards OOD solutions.

Four experimental results :

a) For a simple Bayesian network X_1-> Y->X_2  with linear models and different environments change the Y|X_2 part but not the Y|X_1 part the authors compare REx, IRM and ERM and show that their metric predicts better performance with respect to the causal error and non causal error (measured by presence of just any coefficient on X_2).

b) For Colored MNIST the authors show that when the environments have a varying color-label association , then their metric to test environments is highly correlated with the diversity parameter.

c) Similarly, they compare IRM and REx with different lambda parameters to show that the models with the smallest metric does better in terms of OOD accuracy.

d) For the simpler Linear bayesian network case, they show that for the optimal ERM classifier, only when the environments become identical their metric becomes small. While for IRM, when lambda (regularizer imposing invariance) is large their metric goes to - infty.


Strength:

I feel the metric proposed is novel for an important problem of determining the need for IRM and/or tuning IRM based frameworks where new envrionments have not been seen.

This has been very nicely demonstrated for both simple linear Bayesian networks (also proven for them) and Colored MNIST.

Weaknesses:

1) When authors says we test if the environments are OOD - they dont precisely define it. Only in the colored MNIST experiments this becomes clear. But one has to be very precise about what the authors mean by "testing if environments are OOD" - Does it simply mean one is checking if anything else would do better than ERM in terms of OOD accuracy?

I feel this claim is a bit imprecise. However, due to the colored MNIST demonstration of variation spurious factors being correlated with their metric - I am not very inclined to penalize them in the scores.



2) [My major concern] Another metric to test if a model captures invariance is simply to compute conditional mutual information (CMI) between $e$ ( the environment label) and $Y$ given $\phi(X)$ (the learnt representation). A good place to get a code that computes a good lower bound to this is https://arxiv.org/abs/1906.01824. Their code is public.

Can the authors plot their metric versus this for example for Figure 3.  Because I would like to understand why just directly computing conditional mutual information would be different from the proposed metric.


3) Lots of typos "..when inversing Hessian..", "...unstable inversing..." in Page 8.

I have conditionally given 7 - However some comments, comparison to conditional mutual information would be more useful (Refer to my point 2 in the weakness)

---

> ### Author Response · Authors · 2020-11-17
> **Response**
>
> Thank you for your constructive comments. There are some important points which we hope to clarify and address here and in our revision.
>
> 1. When authors say we test if the environments are OOD - they don't precisely define it. Only in the colored MNIST experiments, this becomes clear. But one has to be very precise about what the authors mean by "testing if environments are OOD" - Does it simply mean one is checking if any-thing else would do better than ERM in terms of OOD accuracy?
>
> **Response:** Yes, "testing if environments are OOD" is checking whether the domains are sufficiently diverse such that anything else would do better than ERM in terms of the OOD accuracy. If the domains are diverse, obtaining high accuracy on a domain requires more domain-specific features. Thus ERM model will employ domain-specific features of training domains which may be very different from the useful features of the unseen test domains. So the ERM may fail to obtain stable performance across all domains. We will clarify this in our revision.
>
> 2.  [My major concern] Another metric to test if a model captures invariance is simply to compute con-ditional mutual information (CMI) between $e$ ( the environment label) and $Y$ given $\phi(X)$ (the learnt representation). A good place to get a code that computes a good lower bound to this is https://arxiv.org/abs/1906.01824. Their code is public.
>
> **Response:** Thank you for the constructive comments. On CMI metric, we went through the paper you pro-posed and implemented the CCIT method in Mukherjee et al. (2020).  Since we do have knowledge of real OOD accuracy in some simple cases, we can test for the consistency between metrics. The results are shown as follows. Numerically, CCIT method does not provide us with an accurate proxy for OOD accuracy.  For ex-ample, the Pearson Coefficient between its metric and real OOD accuracy is 0.377, if we train ERM and REx 100 times each in colored MNIST with 2 training environments. In the same experiment, the Pearson Coefficient between the metric we proposed and OOD accuracy is -0.998. However, we do think that CMI in theory should be a good metric. We therefore explore whether the discrepancy results from issues regarding numeric approximation. To see this, we consider a simple task where we are able to compute conditional mutual Info directly I($e$; $y$ | $\hat y$). With ERM and REx, the Pearson Coefficient between log(real CMI) and OOD accuracy is -0.98 (we apply log function to make the relationship more obvious), better than a -0.909 from our metric in the same setup. We conclude that results from CCIT method might not be present an honest representation of our value of interest. CMI has the potential to be a good metric if the calculation problems can be well-specified and numerically solved, but in such cases as they are not, the metric we proposed would be valuable.
>
> 3. Lots of typos "..when inversing Hessian..", "...unstable inversing..." in Page 8.
>
> **Response:** We are very sorry for that. In the revised version, we have gone through the paper carefully and corrected the typos.
>
>
> ------Reference------
>
> Mukherjee, S., Asnani, H., & Kannan, S. (2020, August). CCMI: Classifier based conditional mutual information estimation. In Uncertainty in Artificial Intelligence (pp. 1083-1093). PMLR.

---

> > ### Comment · AnonReviewer1 · 2020-11-19
> > **Thanks for the experiments and further comments on CMI related issues**
> >
> > " Yes, "testing if environments are OOD" is checking whether the domains are sufficiently diverse such that anything else would do better than ERM in terms of the OOD accuracy. If the domains are diverse, obtaining high accuracy on a domain requires more domain-specific features. Thus ERM model will employ domain-specific features of training domains which may be very different from the useful features of the unseen test domains. So the ERM may fail to obtain stable performance across all domains. We will clarify this in our revision. "
> >
> > $\mathrm{Response}$: I understand this  - but still one needs a formal definition of what it means. Is it possible for the authors to attempt that ? The authors have proposed a test - which is great - however the hypothesis testing problem is not clear ?
> >
> > Regarding experiments for CCMI and CMI:
> >
> > Can you please include this comparison in the paper (supplement) ? If you have done so, thanks. CCMI is a lower bound on mutual information. However it is known that Mutual information when its large numerically, the lower bounds cannot fundamentally approximate it well - this is due to the specific form of Donsker-Varadhan for KL divergence. This argument is from here: https://arxiv.org/abs/1811.04251.
> >
> > My intuition was that since you are doing $I (e; y | \phi(x) )$ - I thought this would be not be numerically high because cardinality of $e$ is small (2 in your experiments).

---

> > > ### Author Response · Authors · 2020-11-24
> > > **Reply**
> > >
> > > Thanks for your response. As for the CMI issue, we will upload a new version including the results and plots of the relationship between CMI and OOD accuracy in the Appendix (we do apologize for failing to upload the latest version earlier). We think an ideal CMI may be useful to address our concerns, but we are not sure since we haven't conducted enough experiments in other tasks like Bayesian network or VLCS. Even in simple tasks as Colored MNIST, the problem of estimation has already emerged (please refer to the appendix). We would suggest more analysis and experiments be carried out to make CMI a better metric.
> > >
> > > **OOD property of domains**
> > >
> > >   Sorry for not to give a rigorous definition of the OOD property of domains. To the best of our knowledge, a mathematical index of the diversity of a set of domains hasn't been proposed in the previous work. In our paper, we circumvent this by setting the goal as maximizing OOD accuracy, i.e. the min accuracy of $S \in \mathcal E_{all}$, given that the subset of domains we can get is $\mathcal E_{tr} \subset \mathcal E_{all}$. Definitely, according to the ``"No Free Lunch" Theorem, we cannot never expect a model to have high OOD accuracy without any assumption of $\mathcal E_{all}$. To give a strict assumption is difficult, but intuitively, we think that the features that are invariant in $\mathcal E_{tr}$ should have a higher probability to also be invariant in $\mathcal E_{all}$. Therefore, we convert the goal of OOD accuracy to the goal of learning invariant features and avoiding features in $\mathcal E_{tr}$. This transformation, as far as we are concerned, is somehow "sufficient", since demanding a model to avoid features that are invariant in $\mathcal E_{tr}$ but varying in $\mathcal E_{all}$ is unrealistic; and is also "necessary", since features that are significantly varied across $\mathcal E_{tr}$ may have unpredictable relation with label in $\mathcal E_{all}$.
> > >
> > >  The metric we proposed is designed to capture whether the features used by the learnt model are invariant. In this case, no matter how the OOD property of domains is defined, if the non-shuffle version and shuffle version of our metric are similar, then to this learnt model, these domains are nearly the same, since it only uses invariant features. Otherwise, we know that the model is using varying features, hence it may have low OOD accuracy.
> > >
> > >  In summary, we argue that the OOD property of domains is a model-based-problem. We conduct experiments in real-world dataset "VLCS" and show how to use the metric step by step to judge the OOD property of the model. The whole process will be included in the new version of the paper, in section 5.3. We hope this may also help!

---

### Decision · Program_Chairs · 2021-01-07
**Final Decision**

**Decision:**

Reject

**Comment:**

The paper considers the question of identifying whether a model is bad from an OOD perspective or certifying that it is good. The reviews agree that there are interesting ideas in the paper, however, lack of sufficient experiments and presentation issues were pointed out which make the paper not ready for acceptance at this stage.